# A structural and dynamic visualization of the interaction between MAP7 and microtubules

Agnes Adler[1,4], Mamata Bangera ®[2,4], J. Wouter Beugelink[3], Salima Bahri[1], Hugo van Ingen ®[1], Carolyn A. Moores ®[2] ✉ & Marc Baldus ®[1] ✉

Microtubules (MTs) are key components of the eukaryotic cytoskeleton and are essential for intracellular organization, organelle trafficking and mitosis. MT tasks depend on binding and interactions with MT-associated proteins (MAPs). MT-associated protein 7 (MAP7) has the unusual ability of both MT binding and activating kinesin-1-mediated cargo transport along MTs. Additionally, the protein is reported to stabilize MTs with its 112 amino-acid long MT-binding domain (MTBD). Here we investigate the structural basis of the interaction of MAP7 MTBD with the MT lattice. Using a combination of solid and solution-state nuclear magnetic resonance (NMR) spectroscopy with electron microscopy, fluorescence anisotropy and isothermal titration calorimetry, we shed light on the binding mode of MAP7 to MTs at an atomic level. Our results show that a combination of interactions between MAP7 and MT lattice extending beyond a single tubulin dimer and including tubulin C-terminal tails contribute to formation of the MAP7-MT complex.

Mitosis, cell migration, and polarization are dependent on microtubules (MTs). An imbalance in their structure or function is associated with human disorders such as ciliopathies, cancer and neurodegeneration[1]. These biopolymers are hollow cylinders of α/β-tubulin heterodimers interacting in a "head-to-tail" and side by side fashion[2,3]. MTs are highly dynamic structures that undergo continuous assembly and disassembly both in vitro and in vivo in a process termed "dynamic instability"[4]. This process is driven by hydrolysis of guanosine 5′-triphosphate (GTP) in the MT lattice[5]. MT-associated proteins (MAPs) interact with MTs, regulate their dynamics, and mediate MT functions in physiological contexts through sophisticated regulation[6]. The roles and mechanisms of MAPs such as MAP1, MAP2, MAP4, Tau, MAP6, DCX, MAP7 and MAP9 have recently been identified and characterized, particularly in the context of MT-rich neurons[6,7]. The less well understood MAP7, also known as E-MAP-115 or Ensconsin, is found to bind and stabilize MTs in axons[8,9], possibly playing a role in the development of axonal branches[10]. It has been implicated in the growth of metaphase spindles in neural stem cells[11] and its upregulation has been observed in different forms of cancer[12–14]. Additionally,

MAP7 is a required cofactor for kinesin-1-driven transport along MTs[15]. Mutations in the gene encoding MAP7 and/or kinesin-1 affect nuclear positioning in myotubes of *Drosophila* embryos as well as mammalian cells leading to muscle defects[16].

Shedding light on the MAP7–MT interaction is important for understanding its physiological function and the wider regulation of molecular processes involving the cytoskeleton. The interaction of MAP7 with MTs is mediated by its 112 residues long MT binding domain (MTBD) (residue 59–170) (Fig. 1a). According to solution-state resonance NMR assignments, the MAP7 MTBD is composed of a long α-helix with a short hinge region comprising residues 84–87[17]. A recent cryo-electron microscopy (cryo-EM) study of the MAP7-MT complex revealed part of the helical MAP7 MTBD[18] bound to the MT (PDBid:7SGS) between the outer protofilament ridge and the inter-protofilament lateral contacts. However, a comprehensive view of the structural organization of the entire MAP7 MTBD on MTs as well as insights into the atomic interactions between MAP7 and the C-terminal tails (CTTs) of α- and β-tubulin, as suggested by previous solid-state NMR experiments[19] are currently missing. In particular, these CTTs are subject to a variety of

[1]NMR Spectroscopy, Bijvoet Center for Biomolecular Research, Utrecht University, Padualaan 8, 3584 CH Utrecht, The Netherlands. [2]Institute of Structural and Molecular Biology, School of Natural Sciences, Birkbeck, University of London, London WC1E 7HX, UK. [3]Structural Biochemistry, Bijvoet Center for Biomolecular Research, Utrecht University, Padualaan 8, Utrecht 3584 CH, The Netherlands. [4]These authors contributed equally: Agnes Adler, Mamata Bangera. ✉e-mail: c.moores@bbk.ac.uk; M.Baldus@uu.nl

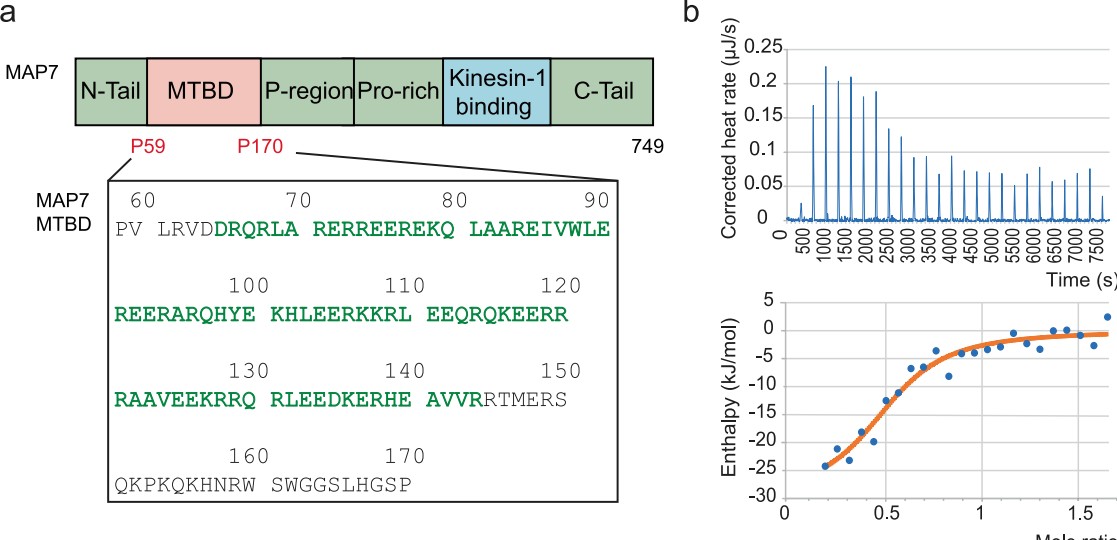

**Fig. 1 | Binding of MAP7 MTBD to MT. a** Top: Schematic representation of full-length MAP7 including the MTBD and the kinesin-1 binding domain. Bottom: Amino-acid sequence of the MAP7 MTBD, residues that are predicted to be α-helical are coloured in green[17]. **b** Representative isothermal titration calorimetry (ITC) data of MAP7 MTBD to Taxol-stabilized MT. The top right picture displays the titration thermogram as heat release after each injection of MAP7 MTBD to the MT. In the lower right picture, the dependence of released heat in each injection is plotted against the ratio of MAP7 MTBD to MT. The experiment was carried out in duplicates and a representative ITC measurement is shown. Source data are provided as a Source Data file.

post-translational modifications (PTMs)[20] affecting the binding of many MAPs to MTs[18,21,22], but little is known about the mechanism(s) by which they contribute to MT binding partner interactions. Although cryo-EM has made great progress in unravelling the structure of MTs and of MAP-MT complexes giving insights into their atomic structure and organisation[6,23], information about the dynamic segments of MAPs - that often contain large intrinsically disordered regions (IDRs), as well as the CTTs - is largely missing. Nuclear magnetic resonance spectroscopy (NMR) has the advantage of not being limited by protein dynamics, making it a suitable tool to study flexible MAP-MT interactions[24,25]. In addition, the use of magic angle spinning (MAS) solid-state NMR allows for the investigation of large biomolecules[26,27]. For example, solid-state NMR has been used to study small ligands[28,29] as well as MAPs and motor proteins binding to MT[19,30–35].

In this study, we establish a comprehensive picture of the binding of MAP7 MTBD to MTs by utilizing a combination of NMR and electron microscopy (EM), supported by fluorescence anisotropy and isothermal titration calorimetry experiments. Our results suggest an interaction with micromolar affinity between an extended α-helix and four tubulin monomers. This significantly expands the previously determined MAP7-MT binding interface, showing that only the N- and C- termini of the MTBD are dynamic and explaining MAP7-promoted MT stabilization. We found that binding of MAP7 MTBD stabilizes the MT lattice via longitudinal interactions along the protofilaments and its binding site is not altered by the presence of MT stabilizing drugs. In addition, we employed solution-state NMR titration experiments using peptides comprising the tubulin CTTs to study the dynamic interactions with MAP7 MTBD at residue-specific level. We identified two important regions in MAP7 MTBD that interact with the CTTs and might be critical for recruitment of MAP7 on the MT lattice. Taken together, our experiments reveal how both strong as well as weak, dynamic protein-protein interactions organize formation of the MAP7-MT complex.

## Results

### MAP7 MTBD binds with micromolar dissociation constant to MTs

To understand MAP7's interaction with MTs, we first analyzed the binding of MAP7's 112 residue long MTBD (Fig. 1a) to MTs using isothermal titration calorimetry (ITC) and solution-state NMR titrations of MTs to labelled MAP7 MTBD. For the ITC experiments, MAP7 MTBD was titrated onto the MTs and a $K_D$ of 0.94 μM ± 0.73 μM could be observed with a stoichiometry of 0.51 ± 0.09 (MAP7 MTBD to tubulin) (Fig. 1b). In addition, we observed strong (90%) signal decrease in N-H-transverse relaxation-optimized spectroscopy (NH-TROSY) experiments at sub-stoichiometric amounts of added tubulin (Supplementary Fig. 1a, b), signifying binding between the labeled MAP7 protein and polymerized tubulin. To obtain residue-specific insight into the dynamic interaction of MAP7 and MT, we subsequently resorted to a combination of EM and NMR experiments.

### MTBD binding along protofilaments stabilizes the lattice

We used EM to directly visualize the effect of the MAP7 MTBD interaction with MTs during polymerization. 5 μM GTP-tubulin (below its critical concentration) was incubated with different concentrations of our MAP7 MTBD construct with no other stabilizing agents, and the resulting MTs were imaged using negative stain EM. We observed an increase in MT number at molar ratios of 0.25:1 and 0.5:1 of MAP7 MTBD:tubulin, whereas no MTs were observed when MAP7 was not included. This demonstrates that MAP7 MTBD promotes MT polymerization (Supplementary Fig. 2). In samples with tubulin and MAP7 MTBD in stoichiometric amounts, bundling of MTs was observed, while at higher concentration ratios of 2:1 for MAP7 MTBD and tubulin, thick-walled short MTs containing an additional layer of protein, potentially composed of tubulin, were observed (Supplementary Fig. 2).

To further understand the MAP7 MTBD−MT interaction, we used cryo-EM for structure determination (Supplementary Fig. 3). We initially polymerized MTs at a ratio of 1:1 (MAP7 MTBD:tubulin), imaged them, and using our established 3D reconstruction pipeline[36,37] that treats the α/β-tubulin dimer as the asymmetric unit of the reconstruction, we obtained a symmetrized 3D reconstruction of MAP7 MTBD bound MT with an overall resolution of 3.7 Å (Supplementary Fig. 4a). The resulting reconstruction showed extra density corresponding to MAP7 MTBD along the protofilaments, but this was only visible at inclusive map thresholds (Supplementary Fig. 5a). To improve the occupancy of MAP7 MTBD on the MT lattice, additional MAP7 MTBD was therefore added to MAP7 MTBD-nucleated MTs

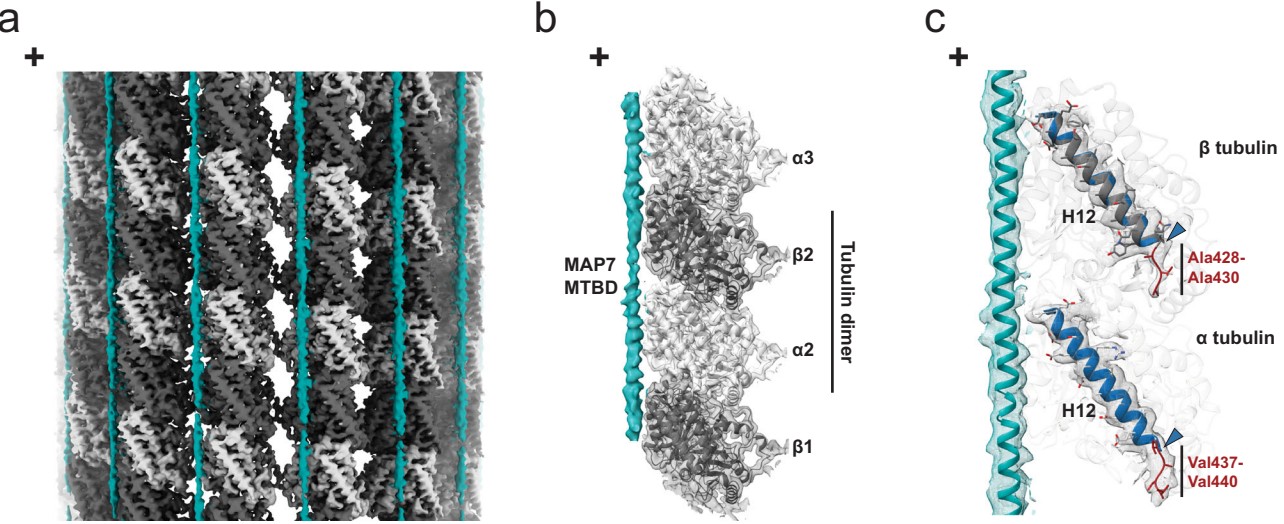

**Fig. 2 | Cryo-EM reveals the binding site of MAP7 and its interactions along MT protofilaments. a** Cryo-EM density (symmetrized reconstruction) of MAP7 MTBD bound MT. Density for the α-tubulin, β-tubulin and MAP7 MTBD is shown in light grey, dark grey and teal respectively. **b** Cryo-EM reconstruction corresponding to a single protofilament (See Methods and Supplementary Fig. 3a for details) showing MAP7 MTBD (teal) bound to four tubulin monomers (alpha tubulin: light grey; beta tubulin: dark grey, models shown in cartoon depiction). Position of tubulins including the central heterodimer have been indicated along the protofilament. **c** Density corresponding to H12 helix in both α- and β-tubulin as well as MAP7 MTBD helix shown in mesh representation. Fitted atomic models for α-tubulin, β-tubulin and bound MAP7 MTBD are depicted in cartoon representation. Residues that form the extension of the C-terminus from H12 helices are indicated in red in both α- and β-tubulin. H12 helices in α- and β-tubulin from atomic model of GDP porcine MT (PDB ID:6DPV, EM map: EMD-7974) superposed on MAP7 MTBD bound tubulin dimer are shown in blue with the corresponding C-terminal ends indicated by blue arrowheads. Minus (-) and plus (+) ends of the MT are indicated in all panels.

adsorbed on the grid before vitrification. Following imaging and processing of these MTs, the symmetrized 3D reconstruction of MAP7 MTBD bound MT had an overall resolution of 3.3 Å and revealed improved MAP7 MTBD density lying along the protofilament, and offset ~20 Å from its centre (Fig. 2a, Supplementary Fig. 4a). α- and β-tubulin were accurately discriminated during data processing, as evidenced by the distinct appearances of their S9-S10 luminal loops (Supplementary Fig. 4a). The helical nature of MAP7 MTBD is also evident from the reconstruction density, albeit at slightly lower local resolution (~4 Å) compared to the MT (Supplementary Fig. 5b). Application of modern cryo-EM processing strategies, including focused classification, particle subtraction and refinement of a single protofilament led to improvement in MAP7 MTBD density for which the helical backbone could be traced[18,38,39] (Fig. 2b, Table 1, Supplementary Fig. 4a). Individual MAP7 MTBD side chains however, could not be discriminated in the helical density, probably due to a number of factors including sub-stoichiometric MT binding and as a result of the dynamic nature of the MT-MTBD interaction. An additional important contributing factor to this is that structural information about the entire MAP7 MTBD - which is predicted to extend across two tubulin dimers - could not be discriminated during our reconstruction process and its precise mode of interaction is therefore obscured through averaging (Supplementary Fig. 5c).

To facilitate interpretation of our reconstruction, we used AlphaFold 2 multimer[40,41] to predict the structure of MAP7 MTBD (residues 59–170) bound to a tubulin dimer. The resulting model predicts that the central region of MAP7 MTBD binds across the tubulin dimer, with helical regions at both N- and C-termini protruding beyond its plus and minus ends. This register of the MAP7 MTBD:tubulin complex is similar to that obtained previously[18] (Supplementary Fig. 5d). The calculated model was fit into the MAP7 MTBD-MT cryo-EM density, and flanking tubulin monomers along the full length of the MTBD were also incorporated for refinement against the electron density map (See Methods and Supplementary Fig. 6a for details). The refined model shows that residues 63–151 in MAP7 MTBD form a long

helix that spans ~13 nm, as was previously predicted by solution-state NMR for the MAP7 MTBD in the absence of tubulin[17]. Its position with respect to the tubulin surface in the predicted model aligned well with the corresponding density in the cryo-EM reconstruction confirming its binding position and helical conformation (Supplementary Fig. 6a). However, since the exact register of the MAP7 MTBD cannot be confirmed based on cryo-EM density alone, we truncated our MAP7 MTBD to include only its helical backbone together with 4 tubulin monomers, including the central α/β-tubulin dimer[18] (Fig. 2b, Supplementary Fig. 6b, c and Table 1).

Docking of this model into the cryo-EM reconstruction also allowed us to identify small extensions of density for both α- and β-tubulin's H12 helix compared to the previously published structure of undecorated GDP porcine MT[42]. Additional C-terminal residues Val437-Val440 for α-tubulin and residues Ala428-Ala430 for β-tubulin were therefore also modelled guided by the density (Table 1). The ordering of these usually disordered residues could be a result of an interaction between the tubulin CTTs and MAP7; however, this interaction is not directly visible in our reconstruction and is presumably not well ordered (Fig. 2c).

To assess our observations about MAP7 MTBD binding in the context of pre-stabilized MTs, which are used elsewhere in this study and in the literature[15,43], we also calculated 3D reconstructions of the same MAP7 MTBD construct bound to Taxol-stabilized MTs on both 13-protofilament and 14-profilament MTs (Supplementary Fig. 4b). Although the overall resolution of this reconstruction is lower than that in the absence of Taxol (possibly due to lower occupancy and/or the presence of Taxol making particle alignment less reliable), these reconstructions showed that the MAP7 MTBD binding site is the same on Taxol MTs as in our MAP7 MTBD-stabilized MT reconstruction (Supplementary Fig. 7). These structures also support the idea that MAP7 binding is not sensitive to MT protofilament number. This same binding site was also observed in 3D reconstruction of various MAP7 constructs (including a similar MTBD construct) bound to peloruside stabilized-MTs[18], demonstrating that MAP7 interacts with MTs in the

**Table 1 | Cryo-EM data collection, refinement and validation statistics**

| | MAP7 stabilized MTs + MAP7 (single proto-filament) (EMDB-19042) (PDB 8RC1) | 13pf Taxol stabilized MTs + MAP7 (single protofilament) (EMDB-19043) | 14pf Taxol stabilized MTs + MAP7 (single protofilament) (EMDB-19044) |
|---|---|---|---|
| Data collection and processing | | | |
| Magnification | 81,000x | 81,000x | 81,000x |
| Voltage (kV) | 300 | 300 | 300 |
| Electron exposure (e–/Å²) | 49.37 (1 and 2) 49.29 (3 and 4) | 49.93 | 49.93 |
| Defocus range (μm) | −0.6 to −2.4 | −0.6 to −2.4 | −0.6 to −2.4 |
| Pixel size (Å) | 1.067 | 1.067 | 1.067 |
| Symmetry imposed | C1 | C1 | C1 |
| Initial particle images (no.) | 186,522 | 217,718 | 217,718 |
| Final particle images (no.) | 39,266 | 40,100 | 39,933 |
| Map resolution (Å) FSC = 0.143 | 3.7 | 3.9 | 4.0 |
| Map resolution range (Å) | 3.5–5.5 | 3.6–6.9 | 3.7–6.3 |
| Refinement | | | |
| Initial model used (PDB code) | Sequence based model from ModelAngelo and AlphaFold2 multimer | | |
| Model resolution (Å) FSC = 0.143 | 3.5 | | |
| Map sharpening $B$ factor (Å²) | N/A | N/A | N/A |
| Model composition | | | |
| Non-hydrogen atoms | 13235 | | |
| Protein residues | 1811 | | |
| Ligands | 6 | | |
| $B$ factors (Å²) | | | |
| Protein | 122.80 | | |
| Ligand | 121.92 | | |
| R.m.s. deviations | | | |
| Bond lengths (Å) | 0.002 | | |
| Bond angles (°) | 0.451 | | |
| Validation | | | |
| MolProbity score | 1.52 | | |
| Clashscore | 5.57 | | |
| Poor rotamers (%) | 1.71 | | |
| Ramachandran plot | | | |
| Favored (%) | 97.83 | | |
| Allowed (%) | 2.17 | | |
| Disallowed (%) | 0.00 | | |

same way independently of how MTs are stabilized (Supplementary Fig. 7; see Discussion for a more extended comparison).

**Solid-state NMR allows direct observation of MT-bound MTBD**
To gain more precise insight into the MAP7 MTBD–MT interaction at atomic resolution, we conducted dipolar and J coupling (scalar)–based magic-angle spinning (MAS) solid-state NMR experiments. These experiments[19,44] provide complementary information about complex biomolecules that comprise both rigid as well as dynamic protein domains and crucially are not limited, as the cryo-EM calculations are, by averaging of tubulin dimers. For our studies, we prepared samples in which [$^{13}$C-$^{15}$N] labelled MAP7 MTBD was mixed with Taxol-stabilized MTs at a 1 to 2 ratio of MAP7 MTBD to tubulin.

The two-dimensional $^{13}$C-$^{1}$H correlation experiments give a fingerprint of the rigid (Fig. 3a, blue) and flexible (Fig. 3a, red) components of the MAP7 MTBD bound to MTs. A comparison of these data sets strongly suggests that the dipolar- and scalar-based ssNMR

experiments probe different domains of MAP7. For further analysis, we made use of our previous solution NMR assignments[17] which cover 97% of CA and 93% of CB resonances as well as 73% of the HA and 69% of the HB assignments (Fig. 3b and Supplementary Table 1) that are included in Fig. 3a as black crosses. Because of spectral overlap, missing assignments in side-chains and chemical shift perturbations, a direct transfer of all solution-state chemical shift assignments to the ssNMR spectra was hindered. Instead, we used in these cases average biological magnetic resonance data bank (BMRB) chemical-shift values (Supplementary Table 1).

Panels (I)-(VII) of Fig. 3a show that Pro, Val, Ala, Asp, Met and Asn residue types are largely dynamic, whereas Gln, Trp and Lys are mostly found to be rigid. Residues Lys154, His167 were additionally identified to be dynamic. Most other residue types revealed ssNMR correlations in both rigid and dynamic protein regimes (*Methods*). In summary, analysis of our 2D ssNMR data sets suggests that flexible residues are more likely to be located at the N- or C-terminus of the protein, while

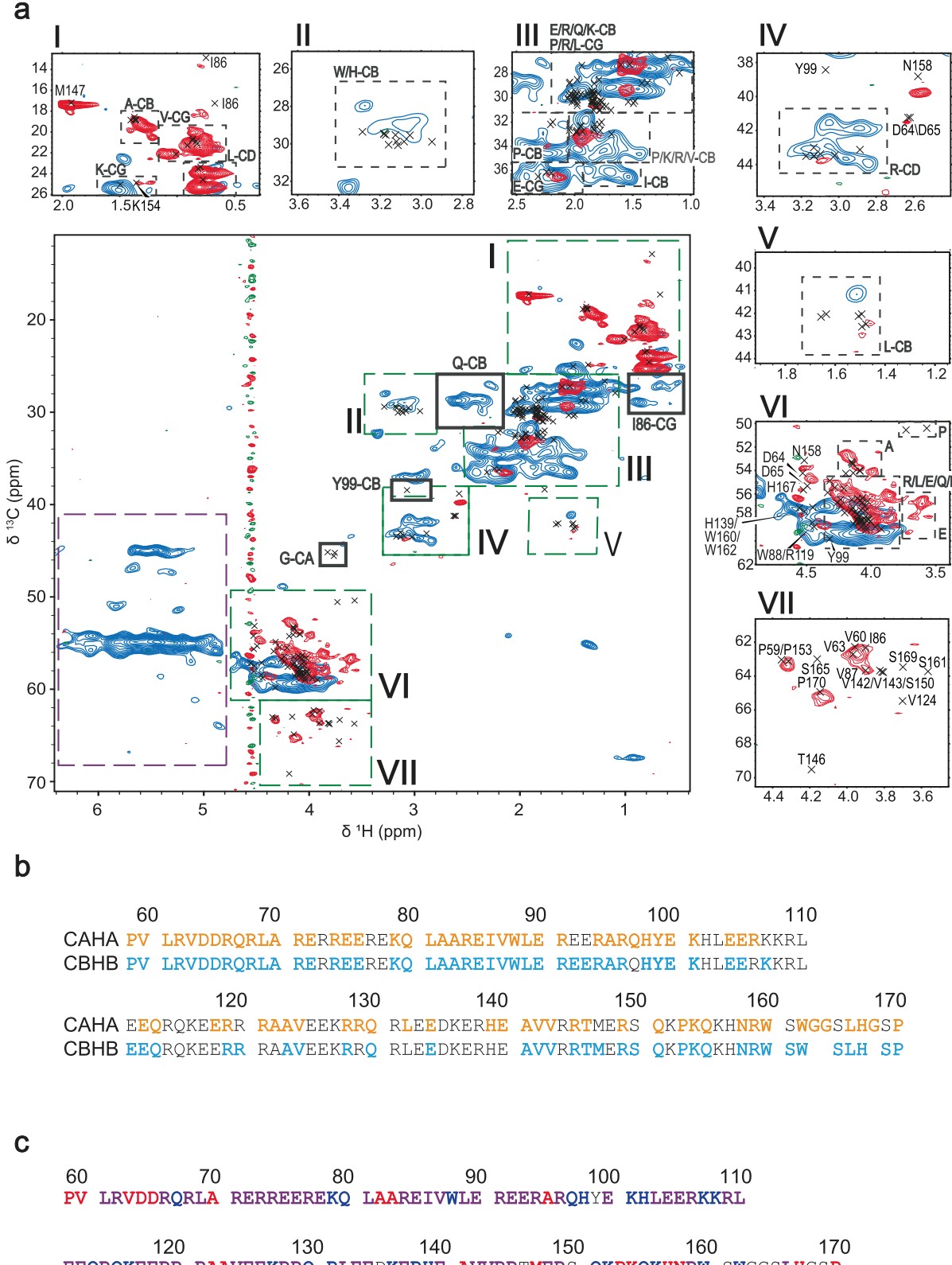

**Fig. 3 | 2D ssNMR of [¹³C-¹⁵N] MAP7 bound to Taxol-stabilized MT. a** Overlay of scalar (flexible) CH (red) and dipolar (rigid) CH (blue) on solution-state assignments of MAP7 MTBD (black crosses). Signal attributed to β-sheet chemical shifts is shown in the purple dashed box, corresponding to CaHa resonances of Ala, Glu, Arg and a fourth correlation which correlates with Val, Ile or Thr (Supplementary Fig. 8c).

Roman numbers and green boxes are used to indicate regions that are enlarged. **b** Solution-state assignments of MAP7 MTBD for CAHA (orange) and CBHB (cyan) resonances **c** Summary of the observations from the 2D NMR spectra. In red are residues that overlay with the scalar CH, in blue those that overlay with the dipolar and in purple those overlaying with both.

rigid residues are mainly found in the α-helical region of MAP7 (Fig. 3c).

Interestingly, additional correlations appeared in our dipolar ssNMR spectra in the proton region between 4.7 and 6 ppm (Fig. 3a, purple dashed box), which cannot be explained by our solution NMR data or a protein species that comprises α-helical or random-coil conformations. Instead, these signals exhibit β-strand character (*vide infra*). Notably, aggregation of MAPs with MT stabilizing function has been observed for MAP1b, MAP2, TPPP and Tau[45–47].

To validate these findings and to obtain further residue-specific information we conducted three-dimensional CCH solid-state NMR spectra (Fig. 4, Supplementary Fig. 8a) using both dipolar as well as scalar-based transfer steps.

The 3D scalar spectrum (probing dynamic MAP7 residues in the complex) showed good agreement with the solution-state assignments. Here, we were able to assign several residues of MAP7 MTBD in the MT-bound state (Fig. 4a, c). For the Pro 59- Asp 64 stretch, Leu 69, Glu 148, Pro 153, Gln 155, Asn 158, Arg 159 and Pro 170 resonances for both backbone and sidechains could be identified (Figs. 4a and 5a), while for Ala 70, Ala 82, Ile 86, Thr 146, Met 147, Lys 154 and Leu 166 only certain side-chain resonances were found (*Methods*). Intriguingly, all of these residues are located at the C- and N-terminus of MAP7 MTBD (Fig. 5a), in line with weak or no binding to MT for MAP7 residues Pro 59-Ala 70 and Glu 148-Pro 170. In addition, several resonances corresponding to residue types could be identified in the scalar CCH 3D, namely, one Glu, Ser, Gln, Leu and two additional Arg (Fig. 4a). Furthermore, we normalized the number of resonances of each residue type present in the aforementioned terminal domains to its overall abundance in the entire sequence. Taking the information gained from the experimental signal intensity together with the observed residues we find our scalar based experiments to be in reasonable agreement with the MAP7 regions Pro 59-Ala 70 and Glu 148-Pro 170 (Fig. 5b, Supplementary Fig. 8b, *Methods*).

For the dipolar spectra (Fig. 4b, d, Supplementary Fig. 8c), we were able to assign Ala, Glu, Lys, Leu, Met, Gln, Arg, Ser and Val residue types with distinct chemical-shift signatures (Supplementary Table 2). These MAP7 MTBD residues are more abundant in the central α-helical region as seen in the prediction model fit into EM density and extend to approximately residue 150 (Fig. 5a, c). Indeed, these residue types exhibit α-helical shifts, underlining the α-helical secondary fold of the bound part of the MAP7 MTBD (Fig. 5d). It is interesting that Gln CBHB is an exception to the rule, as the chemical shift is more typical of random coil shifts than the average Gln CBHB chemical shift in solution. However, we identified only one Gln CBHB in the 3D CCH (Supplementary Table 2), which could be located in a region at the beginning or end of the helical part of MAP7 MTBD and was hence separated enough from the overlapping Gln resonances to be assigned (e.g., Gln 80 or Gln 151).

We note that the aforementioned correlations are the ones that were unambiguously assignable. However, there are several spectral regions in which overlap precludes the assignment of individual resonances but still allows for a residue-type analysis. Similar to the scalar case, we calculated the relative abundance of these resonance-type specific regions by dividing the integral of the resonance region in the CH projection of the 3D CCH spectrum by the average of the integral of peaks corresponding to one assignment only (Fig. 5c). The residue abundance is in good agreement with the number of these residues in the region Ala 70-Glu 148 (Fig. 5c, right and Supplementary Fig. 8b), corresponding to the domain that we previously found absent in the scalar spectrum (Fig. 5a). This notion is confirmed by the presence of all expected Ala, Arg, Val, His, Gln and Trp side-chain resonances. Nonetheless, fewer Lys resonances were observed and none for Asp. The discrepancies may be due the fact that some of the residues exhibit a certain degree of motion which reduces dipolar transfer and cross-peak intensities in our dipolar ssNMR spectra.

In addition to the ssNMR data presented above, we also obtained sequential assignments from dipolar 3D CCH experiments recorded with a longer Radiofrequency driven dipolar recoupling (RFDR[48]) mixing time (3.4 ms) (Fig. 4e). By comparing this 3D data set to results of the dipolar 3D experiment with shorter mixing time we could connect two residue pairs. Firstly, a sequential Ala-Leu contact would only be consistent with pairs 69–70 and 81–82. Because we identified Leu 69 in the scalar 3D, we tentatively assigned this correlation to Leu 81-Ala 82. Moreover, we observed sequential transfer from Ser to Arg which must stem from residues Arg 149 to Ser 150. Therefore, it seems likely that the rigid region extends up to approximately residue Ser 150, with possibly weaker binding starting from Thr 146 (Figs. 4e, 5c). The assigned resonances can be found in the supplementary table (Supplementary Table 2).

Similar to the analysis of the α-helical region, we also attempted a residue-specific analysis of the putative β-strand region. By verifying connections in the CAHA region of the dipolar spectrum corresponding to the aforementioned β-sheet resonances, we tentatively assigned residues that best agree with Ala 83, Arg 84, Glu 85, Ile 86 and Val 87 (Supplementary Fig. 8b and Methods). These residues would be in line with the aggregation propensity for MAP7 evaluated by AGGRESCAN, which identifies residues 84–89 to be aggregation-prone[49]. Interestingly this region corresponds to the hinge, that was observed in the free MAP7 MTBD α-helix (residue 84–87)[17]. The rest of the aggregate could not be observed in the NMR spectra, possibly due to static conformational heterogeneity or intermediate exchange dynamics.

Taken together, our spectral analysis suggests a picture in which certain MAP7 residues are either flexible or rigid as depicted in Fig. 5a. We conclude that dynamics are prevalent in the N-terminus before residue 70 and in the C-terminus after residue 148.

## Bipartite interaction of MAP7 MTBD with tubulin tails

Previous work hinted at the importance of the tubulin CTTs in binding of MAP7 to MTs as indicated by a reduction in the scalar ssNMR signal of the CTTs[19,34]. To examine the binding of MAP7 MTBD to the tubulin CTTs, we designed peptides comprising residues Ser 439-Tyr 451 (439-SVEGEGEEEGEEY-451) and Asp 427-Ala 444 (427-DATAEEEEDFGEEAEEEA-444) of HeLa S3 α- and β-tubulin, respectively. By titrating these to isotopically labelled MAP7 MTBD in solution (Fig. 6a), we were able to observe chemical-shift changes (Fig. 6b). Interestingly, we identified two main regions (82-AAREIVW-88) and (141-AVVRRT-146) that are affected by the presence of the CTTs peptides, while the latter region seems to be less modulated by the β-tubulin tail (Fig. 6b). Notably, we also detected other residues with lower sensitivity due to the presence of the CTT peptides, i.e., Asp 65, Glu 75, Arg 106, Ala 122 and Ser 150. We determined the binding affinity for the α-CTT with fluorescence anisotropy by attaching NHS-fluorescein to the peptide (Fig. 6c). The experiments resulted in a dissociation constant ($K_D$) of 91.3 ± 17.5 μM. This binding affinity seems therefore to be 90 times weaker than the micromolar interaction observed for binding of MAP7 MTBD to entire MTs (Fig. 1b). The electrostatic character of the CTT–MAP7 interaction was validated by NMR measurements of MAP7 MTBD with 10-fold α-CTT at different salt concentrations. A reduction of chemical shift perturbations due to salt increase could be observed, leading to a disappearance of any relevant chemical shift perturbations at 500 mM salt (Fig. 6d). The dynamic binding behaviour was also verified by previous total internal reflection fluorescence experiments showing a 4-fold reduction in the binding of the MAP7 MTBD to MTs upon removal of the CTTs via subtilisin cleavage[18].

Our findings, in conjunction with the highly negative charge of the CTTs, suggest that the charged residues of the interacting stretches play a crucial role in an electrostatic binding event between MAP7 and MTs. Furthermore, the electrostatic character of the interaction is

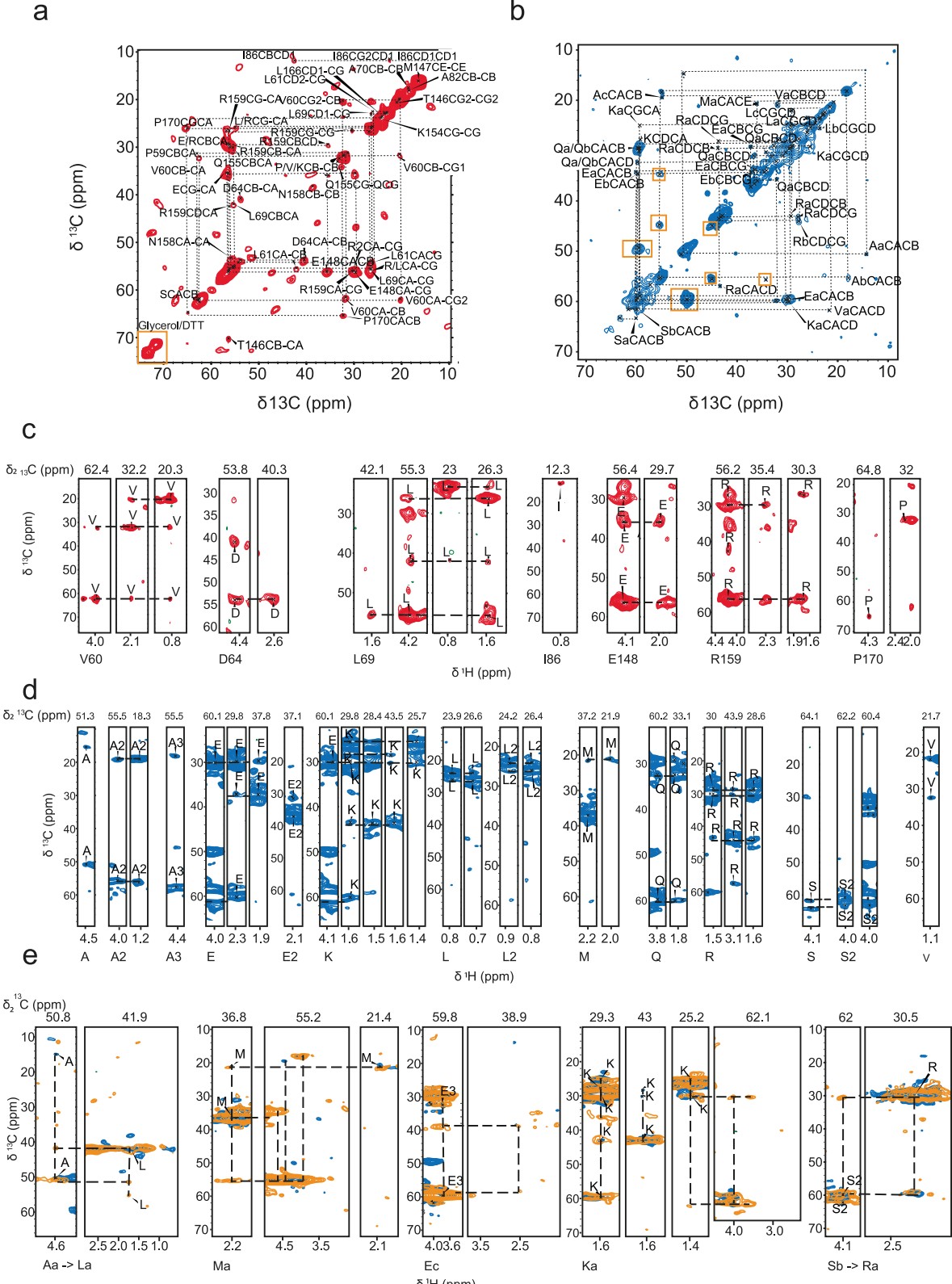

**Fig. 4 | 3D ssNMR of [¹³C-¹⁵N]-MAP7 bound to Taxol-stabilized MT. a** CC-plane of 3D 11 ms DIPSI with assigned resonances. Diagonal signals in the yellow box stem from buffer components. **b** CC-plane of 3D 1.7 ms RFDR with assigned resonances. Yellow boxes indicate correlations stemming from β-sheet signals. **c** Assigned strips of the scalar 3D CCH. **d** Assigned strips of the dipolar 3D CCH. **e** Assigned strips in the dipolar spectrum for 3.4 ms RFDR mixing time (yellow) with sequential connectivity compared to 1.7 ms RFDR (blue). Amino-acid (but not residue)-specific chemical shifts are denoted by letters.

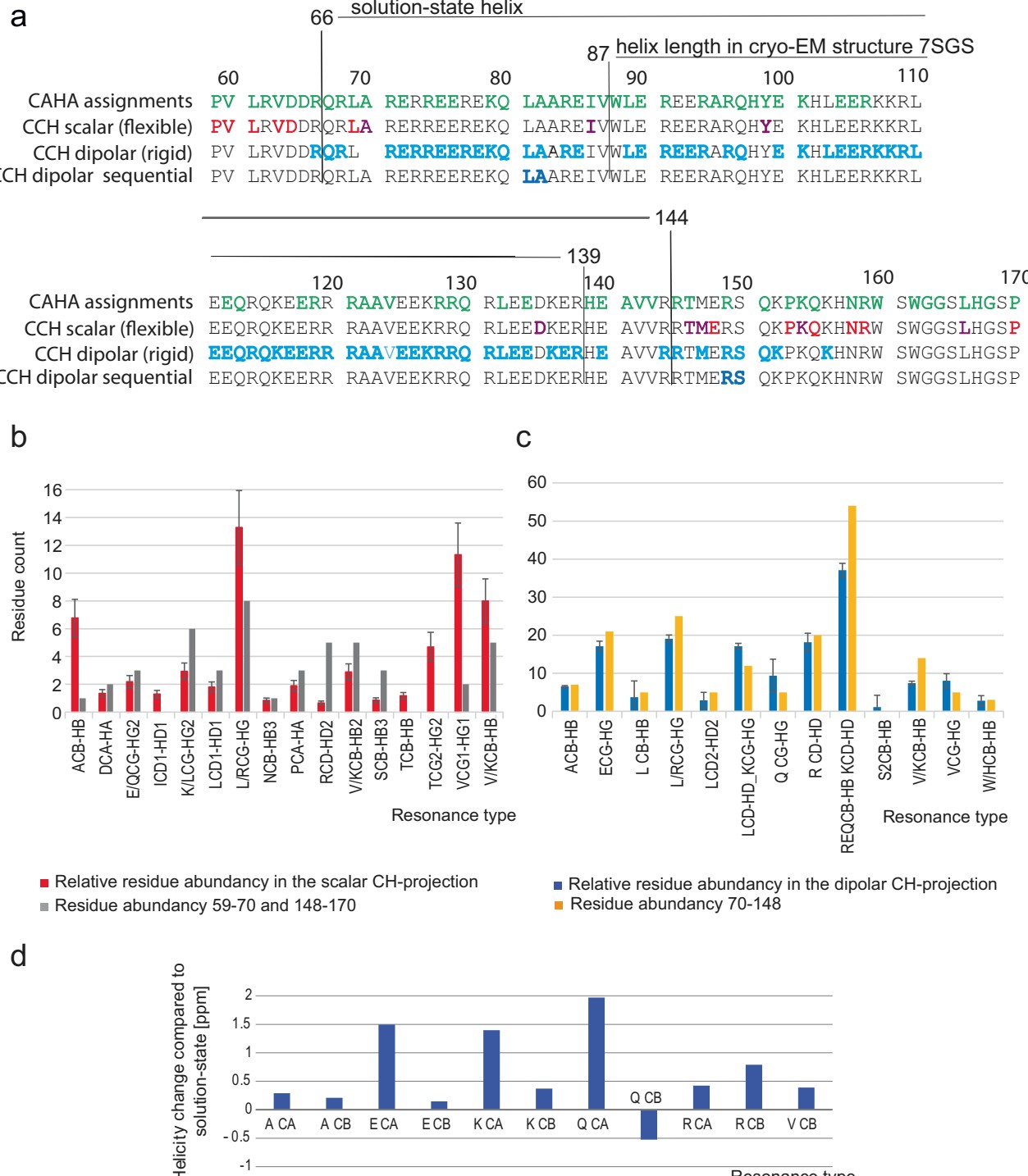

**Fig. 5 | Analysis of 3D CCH ssNMR experiments. a** Distribution of assigned residues indicated on the MAP7 MTBD sequence. Top: CAHA assignments in solution (green), 2nd row: assigned residues in the CCH scalar 3D (red for several resonances, magenta for only side-chain assignments), 3rd row: residue types found in dipolar CCH (blue), bottom row: residue pairs identified in sequential dipolar CCH. The length of the helix found in solution and in the previous cryo-EM study are indicated by black lines. Comparison of predicted (residue count) and experimental signal intensities (integrated intensity) in scalar CCH (**b**) and dipolar CCH (**c**) regions. The integrated intensities were divided by the average signal intensity of one resonance only. Error bars were derived from experimental signal-to-noise ratio (SNR) which is determined by the ratio of the average height of the NMR peak to the standard deviation of the noise height in the baseline. **d** Comparison of helicity of solution-state assignment average for residues A70-E148 to the average chemical shift helicity of the assigned same residue type in the dipolar CCH spectrum in ppm (for further details, see also Methods). Source data are provided as a Source Data file.

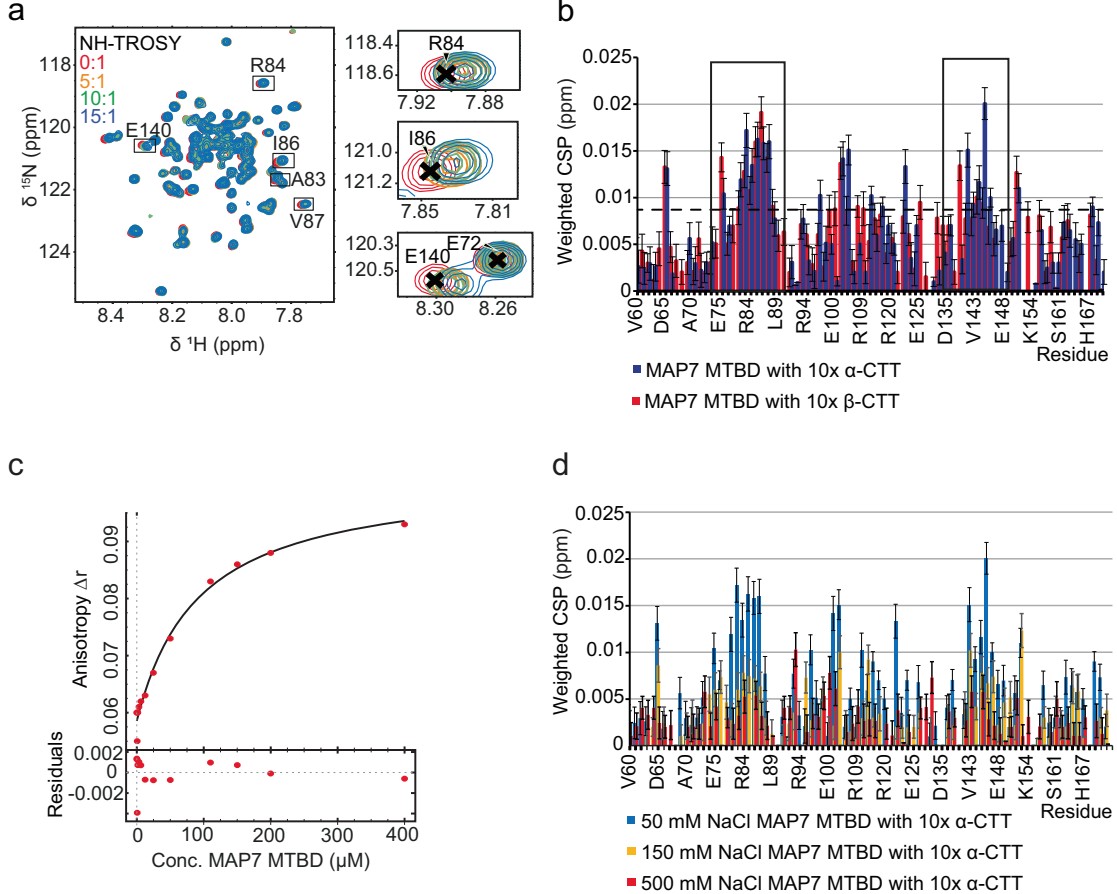

**Fig. 6 | Titration of peptides comprising the MT carboxy-terminal peptides to [¹³C-¹⁵N]-MAP7 MTBD. a** NH-TROSYs of the titration of increasing concentrations of α-CTT to MAP7 MTBD in solution-state NMR. **b** Chemical shift perturbation (CSP) upon α- and β-CTT binding. Stretches with significant CSPs are marked with a box. The black dotted line indicates the mean plus one standard deviation. **c** Fluorescence anisotropy with fluorescein labelled α-CTT gives a $K_D$ of 91.3 μM ± 17.5 for the interaction with the MAP7 MTBD. **d** CSP in labelled MAP7 MTBD upon addition of 10x α-CTT at different salt concentrations. CSP were calculated as described in ref. 83. In (**b**, **d**) the error bars were calculated using standard error-propagation techniques. The error in the peak position was estimated by dividing the average line-width (in ppm) of each peak along the ¹H and ¹⁵N dimension by the signal-to-noise ratio. Source data are provided as a Source Data file.

underlined by the observation that the FGE and EGE repeats of the CTTs are the binding residues on the MT side and therefore negatively charged residues might interact with the charged MAP7 residues[19].

## Discussion

MAP7 mediates recruitment of kinesin-1 to the MT lattice and plays a crucial role in the organization and transport of cellular cargo via MT-based motors, in organelle movement and spindle segregation[15,50,51]. The interaction of MAP7 with MTs is associated not only with kinesin-1-mediated transport in cells but also with MT remodelling[8]. As a result, studying this interaction will aid in understanding numerous cellular processes, which are linked to cancer-associated metastatic growth and neurodegenerative diseases[12,43,52,53].

Initially, we studied the interaction between MAP7 MTBD and MTs by fluorescence anisotropy and isothermal titration calorimetry experiments. Our ITC data revealed a Kd in the uM regime. Earlier findings using multiple MAP7 constructs and Total Internal Reflection Fluorescence microscopy data suggested MT binding with a range of affinities[18,43]. Hence, different MAP7 subregions are likely to contribute uniquely to complex formation. For this reason, we subsequently utilized a combination of NMR and EM, allowing a more complete visualization of this interaction than has previously been possible. Despite the highly repetitive distribution of Arg, Glu, Lys residues, we were able to acquire 2D and 3D solid-state NMR (ssNMR)

spectra of labelled MAP7 MT-binding domain (MTBD) in complex with Taxol-stabilized MTs with remarkable spectral resolution. From these experiments, and as supported by a combination of cryo-EM 3D reconstructions and AlphaFold2 prediction, we show that MAP7 MTBD adopts an extended helical conformation when bound to MTs.

We used exclusively a MAP7 MTBD construct (residue 59–170) in our study, but the MT binding site we visualized is the same as was observed in a previous cryo-EM study using a range of MAP7 constructs (full length MAP7, residues 60–170, residues 83–134[18]). In that study, the MTs used were stabilized by the drug peloruside, which has been shown to bind on the outer surface of MTs only ~15 Å from the MAP7 binding site[54]. The observation that MAP7 binds at the same MT site in all reconstructions to date – either in the absence of stabilizing drug or on MTs stabilized by Taxol or peloruside - collectively shows that MAP7 MT binding is overall not influenced by the mechanism by which MTs are nucleated and stabilized. Comparison of these structures thereby provides important cross-validation between the different experiments conducted and yields valuable information about the robustness of the MAP7 MTBD-MT interaction.

FL MAP7 has a ~10-fold higher affinity for MTs compared to shorter constructs, including our own (Fig. 1b) which likely contributes to the improved occupancy of MAP7 density in the reconstruction of FL MAP7[18]. However, this also shows that not-yet-visualized regions outside the MTBD enhance MT binding by longer MAP7 constructs. In

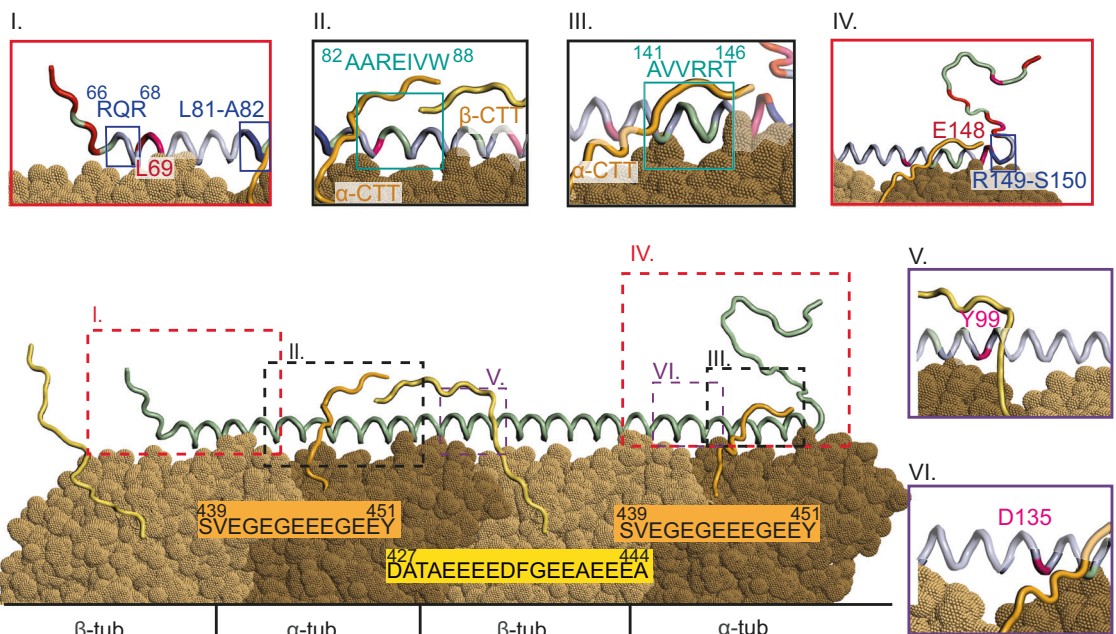

**Fig. 7 | Model of MAP7 MTBD binding to MT based on the cryo-EM and NMR data.** MAP7 MTBD forms a 130 Å long helix from around residue R66 to residue E148 and binds MTs along the protofilament as confirmed by cryo-EM. On the N-terminal side backbone resonances with fast dynamics are only found prior to R66, with the exception of L69 and on the C-terminal side after E148 (red coloured, box I, IV). On the C-terminal side (IV) a region with flexible and rigid residues is observed from T146 until S150. For T146 and M147 only side chain resonances are observed (pink). Box II and III. show the interactions of MAP7 MTBD with the tubulin CTT. The residue stretches undergoing significant chemical shift pertur-bations are indicated by a teal square (box II.: A82-W88; box III.: A141-T146). I86 has flexible side-chains (pink, box II). In both cases, the CTTs are long enough to reach the identified MAP7 regions and also the residues of MAP7 on the neighbouring protofilament. Panels V and VI show residues with intermediate side-chain dynamics (pink) located between tubulins.

addition to the lower MT binding affinity of truncated MAP7 proteins, averaging of αβ-tubulin dimer asymmetric units in cryo-EM recon-structions blurs information about individual side chain interactions between MAP7 MTBD and the MT surface (reviewed recently in ref. 55). In silico modelling of the interaction between the helical MAP7 MTBD and tubulin was therefore used to provide critical, albeit indirect, information about the register of the bound MAP7 MTBD on MTs, similar to the model obtained previously[18]. Crucially, however, in our NMR experiments, determination of the involvement of individual amino acids at the binding interface does not rely on the 80 Å repeat of the underlying tubulin. These experiments thereby provided direct evidence of MT binding along the majority of the ~130 Å helical MAP7 MTBD.

Our ssNMR experiments identified the rigid and flexible parts of MAP7 MTBD, thereby also establishing the helix boundaries and pin-pointing specific interacting regions of the MAP7-MT interaction. We detected fast backbone dynamics in residues N-terminal of Arg66 (with the exception of Leu69) and after Glu148 on the C-terminal side (Fig. 7I, IV), with the latter region (Thr146 until Ser150) comprising both flexible and rigid residues and only side chain resonances observed for Thr146 and Met147. Taken together, these results suggest that the MT binding interface of MAP7 extends beyond a single α/β-tubulin heterodimer. The register depicted in Fig. 7 is consistent with the NMR observation of more flexible residues in the clefts between the tubulin subunits (Fig. 7V, VI) and with previous[18] and our current cryo-EM models as well as AlphaFold 2 multimer predictions. The resulting bridge between two tubulin dimers is predicted to support MT stabilization, as was shown when MTs were polymerized in sub-stoichiometric concentration ratios of 0.5:1 (MAP7 MTBD:tubulin) in vitro (Supplementary Fig. 2), and is consistent with its observed stabilizing activity at axonal branches[56].

In addition to our earlier ssNMR results that revealed the binding site of MAP7 to the MT tails[19], solution-state NMR here provided

insights into the dynamic interaction between tubulin's carboxy-terminal tails (CTTs) and the MAP7 MTBD (Fig. 7II, III). We identified two regions that displayed significant chemical shift changes. The first (residues 82–88) was found to interact with both the α- and β-CTTs, while the second region (residues 141–146) preferably interacted with the α-CTT. This interaction is further supported by the partial ordering of the tubulin C-terminal extensions of the H12 helix that were observed in the cryo-EM reconstructions (Fig. 2c). Together with pre-vious studies that revealed the binding of MAP7 to the EGE and FGE motifs in MT CTTs[19], we now obtain an atomic-level understanding of the interaction between CTTs and MAP7. The long-range electrostatic interaction between the CTTs and the protein may help to attract MAPs from the cytoplasm to the MTs. Once binding has occurred, the relatively weak interaction between CTTs and MAP7 might contribute to stabilizing the MAP7-MT protofilament complex. This notion is also in line with observed weaker binding affinities of MAP7 to subtilisin-treated MTs[18].

Some ambiguities, such as whether the Arg residues in MAP7 involved in CTT binding are rigidified by the tail interaction or by the protofilament binding, cannot be clarified based on our ssNMR data. In addition, we cannot exclude the possibility that the CTTs may interact with the distal MAP7 bound to the neighbouring protofila-ment. Further studies with shorter MAP7 constructs might help to evaluate the importance of the N- and C-terminal interacting parts for MT stabilization. On the other hand, our approach could also be extended to study longer MAP constructs that are known to contain larger IDP (intrinsically disordered protein) regions[57], including MAP7 constructs containing the kinesin-1 binding domain. Finally, investi-gating post-translational modifications (PTMs)[58] of MT tails may pro-vide insights into the importance of the "tubulin code" for binding affinities and binding rates leading to a more comprehensive under-standing of these modifications and their implications in health and disease.

## Methods

### MAP7 MTBD expression and purification

The cDNA encoding the MTBD (residues 59–170) of MAP7 from *Homo sapiens* with an N-terminal His-Tag and Maltose binding protein (MBP) linked via a thrombin cleavage site (His-MBP-MAP7 MTBD), cloned into the pLICHIS vector was expressed in *Escherichia coli Rosetta 2 (DE3)* cells and purified with immobilized metal affinity chromatography, cation exchange chromatography, thrombin cleavage and ammonium sulphate precipitation[17].

### Multi-angle light scattering

Size-exclusion chromatography of the MAP7 MTBD was performed on a Shimadzu UFLC with a Superdex 75 Increase column (Cytiva), calibrated with 40 mM phosphate buffer, 150 mM NaCl, 1 mM dithiothreitol (DTT), pH 6.5, and coupled to a miniDAWN TREOS multi-angle light scattering detector (Wyatt) and a RID-10 A differential refractive index monitor (Shimadzu). 2 mg/mL ovalbumin was used as a standard to perform signal alignment and normalization. 0.8, 1.6, and 3.2 mg/mL MAP7 MTBD were injected, and collected data was analyzed using ASTRA6 software (Wyatt).

### Isothermal titration calorimetry assay

Lyophilized tubulin (Cytoskeleton) was solubilized in BRB80 buffer (80 mM PIPES, 2 mM $MgCl_2$, 1 mM EGTA, pH 6.5, 1 mM $NaN_3$, 1 mM DTT, pH 6.5 supplemented with protease inhibitor (Sigma-Aldrich, cOmplete EDTA-free)), to a final concentration of 2 mg/mL. Then 1 mM Guanosine-5'-triphosphate (GTP) was added and incubation took place for 15 min at 30 °C. In the following, 20 μM paclitaxel (Taxol, SIGMA) was used to stabilize the MT and incubation took place for another 15 min at 30 °C. The MT were spun down at 180000 g (Beckman TLA-55 rotor) for 30 min at 30 °C and the pellet was resuspended in warm BRB80 buffer with 20 μM paclitaxel. The BRB80 buffer was then exchanged to warm NMR buffer with 1 mM β-mercaptoethanol (β-ME) instead of DTT and 1 mM GTP with a PD-10 desalting column.

Isothermal titration calorimetry was performed with the NanoITC (Waters LLC, New Castle, DE, United States) to determine the interaction between Taxol-stabilized MT and MAP7 MTBD in NMR buffer with 1 mM β-ME instead of DTT and supplemented with 1 mM GTP. Samples were degassed before performing the experiment. 150 μM MAP7 MTBD was titrated in 2 μL steps to 164 μL of 15 μM Taxol-stabilized MT. The protein was titrated at a rate of 2 μl/300 s with a stirring rate of 300 rpm. Experiments were performed at 37 °C. Control experiments were performed with buffer titration. The dissociation constant ($K_D$) value was calculated using the Nano Analyse Software (Waters LLC).

### Negative stain sample preparation and EM

Porcine brain tubulin (Cytoskeleton, Cat No. T240) (5 μM) was incubated with MAP7 MTBD in BRB80 buffer (80 mM PIPES, 2 mM $MgCl_2$, 1 mM EGTA, pH 6.8) and 2 mM GTP at indicated molar concentration ratios in a water bath maintained at 37 °C for an hour. 3 μL of each sample was applied to glow discharged continuous carbon film coated, 400 mesh copper grids (Pacific Grid Tech) and blotted using a Whatman filter paper, Grade No. 1 after 30 s. Staining solution (2% uranyl acetate in water) was then added to the grid, blotted off after 30 s and the grids were air dried. Images were recorded using Digital Micrograph™ software (Gatan) on a Tecnai T12 transmission electron microscope (Thermo Fisher Scientific) operating at 120 kV with a US4000 4 K × 4 K CCD camera (Gatan) at 3200x (overview) and 42000x (high) magnifications corresponding to pixel sizes of 13.5 nm and 2.5 Å respectively.

### Cryo-EM sample preparation

For preparation of MAP7 MTBD stabilized MTs, 5 μM porcine brain tubulin (Cytoskeleton, Cat No. T240) and 5 μM MAP7 MTBD were mixed with 2 mM GTP (final concentration) in BRB80 buffer (80 mM PIPES pH 6.5, 2 mM $MgCl_2$ and 1 mM EGTA) and incubated on ice for 5 min. The mixture was then incubated in a water bath maintained at 37 °C for an hour. C-Flat 2/2 Holey Carbon grids (Protochips) were treated in air for 40 s at 0.2–0.3 Torr using a Harrick Plasma Cleaner. 3 μl of the above sample was applied to and incubated on the surface-treated grid for 30 s at room temperature following which, excess sample was wicked off. For higher MAP7 occupancy and structure determination, 3.5 μL of 10 μM or 25 μM MAP7 MTBD was then immediately added to the grid. The grid was then mounted in a humidified Vitrobot Mark IV (Thermo Fisher Scientific) chamber pre-set to a temperature of 25 °C and humidity of 100%. After incubation for 60 s, excess liquid was blotted out and the grid was plunge frozen in liquid ethane.

To prepare Taxol stabilized MTs, 100 μM porcine brain tubulin was mixed with 2 mM GTP (final concentration) in BRB80 buffer and incubated on ice for 5 min. The mixture was then incubated in a water bath maintained at 37 °C for 1 h followed by addition of 200 μM Taxol (Paclitaxel, Sigma-Aldrich Cat No. 580555) and further incubated for 3 h. Taxol stabilized microtubules were pelleted at 278088.3 g in a TLA 100 rotor using a tabletop ultracentrifuge followed by resuspension in BRB80 buffer containing 200 μM Taxol. 10 μM of polymerized MTs were incubated with MAP7 MTBD (desalted using Zeba spin desalting columns) in a ratio of 1:3 in a water bath set at 30 °C for 10 min. 4 μl of this sample was applied to surface-treated C-Flat 2/2 Holey Carbon grids mounted in a humidified Vitrobot chamber pre-set to a temperature of 25 °C and humidity of 100 % for 30 s following which excess liquid was blotted out and the grid was plunge frozen in liquid ethane.

### Cryo-EM data collection

Data were collected using a Titan Krios D3771 microscope (Thermo Fisher Scientific) operating at an accelerating voltage of 300 kV attached to a BioQuantum K3 direct electron detector (Gatan) and post-column GIF energy filter (Gatan) with a slit width of 20 eV. The sample was imaged using the automated EPU software at a nominal magnification of 81,000x resulting in a pixel size of 1.067 Å and with defocus range of −0.6 to −2.4 μm. Data collection statistics for MAP7 MTBD and Taxol stabilized MTs are provided in Table 1.

### Cryo-EM image processing

Initial processing involved manual inspection of data for presence of contaminating ice and filament quality and poor-quality movies were discarded. All subsequent image processing steps were carried out using RELION v3.1[59,60] and the customized Microtubule RELION-based Pipeline (MiRP)[36,37] (Supplementary Fig. 4a). Beam induced motion of particles in the movies were corrected using inbuilt function of MotionCor2[61] in RELION. CTF estimation was performed on dose-weighted and motion corrected summed images using CTFFIND 4.1[62] within RELION's GUI. Particle picking was performed manually using RELION's helical picker[63] and particles were extracted using a box size of 568 pixels and overlapping inter-box distance of 82 Å. The next steps of protofilament number segregation and Euler angle alignments were carried out using 2x binned particles and references with the help of MiRP v2 scripts[37] integrated into RELION v3.1. MT segments with different protofilament numbers were segregated by supervised 3D classification using references for 11 to 16 protofilament containing MTs. 14 protofilament MTs formed the major fraction of the population and were re-extracted with a box size of 448 pixels for further processing. Rotational angle and X/Y coordinate fitting and shift assignments were carried out using 3D classification against a 15 Å low pass filtered reference (EMD-7973) with enhanced pixels for S9-S10 and H1-S2 loop regions in tubulin[64]. Seam checking was carried out by supervised 3D classification using 28 rotated and shifted references. Finally, a symmetrized 3D reconstruction was obtained with a 3D auto-refine step using unbinned aligned particles and a 10 Å lowpass filtered

reference (Table 1). Per particle CTF refinement and Bayesian polishing were then carried out in RELION. At this stage, aligned polished particles from 4 datasets (corresponding to 10 μM and 25 μM MAP7 externally applied) were combined and used for the final round of 3D reconstruction with (application of helical symmetry with rise = −25° and twist = 9 Å) and without symmetrization (C1 symmetry). The final displayed reconstruction in Fig. 2a and Supplementary Fig. 5b corresponds to the symmetrized reconstruction that was sharpened using local resolution in RELION. After C1 refinement, the corresponding particles were expanded 14-fold using helical symmetry and subjected to another round of 3D refinement in RELION. The 'good protofilament' opposite the seam was then subtracted from the particles in RELION using a soft mask around a single protofilament. The subtracted particles were then locally refined in cryoSPARC[65] and local resolution estimated in RELION. This reconstruction is shown in Fig. 2b, c and Supplementary Fig. 7 (EMD-19042).

Data processing was carried out in the same manner as above for both 13 and 14-protofilament containing Taxol stabilized MTs (Supplementary Fig. 4b). The 3D reconstructions depicted in Supplementary Fig. 7 (EMD-19043 and EMD-19044) correspond to single protofilament 3D reconstructions using subtracted particles focused on the protofilament opposite the seam from a refined symmetry-expanded dataset.

### AlphaFold2 multimer prediction

Protein sequences for porcine brain alpha-1A tubulin (UniPROT ID: P02550), beta-2B tubulin (UniPROT ID: P02554) and MAP7 MTBD (Residues 60–170 from UniPROT ID: Q14244) were used as inputs for locally installed AlphaFold2 multimer[40]. Out of the 25 models obtained, the best model assessed by pTM+ipTM score was used for fitting and analyses.

### Model generation and refinement

Protein sequences for porcine brain tubulin: porcine brain alpha-1A tubulin (UniPROT ID: P02550), beta-2B tubulin (UniPROT ID: P02554) and the symmetrized electron density map with a mask around the tubulin dimer in the centre of the protofilament opposite the seam were provided as inputs to locally installed ModelAngelo[66]. The model obtained from ModelAngelo was further checked and refined manually in *Coot*[67] followed by real-space refinement in PHENIX[68–70] (Supplementary Fig. 6a). The alpha tubulin chain from the AlphaFold2 multimer model obtained as described above was superimposed on the alpha tubulin chain of the built tubulin dimer. After superposition, the chains corresponding to tubulin were removed from the AlphaFold2 model and only the chain corresponding to MAP7 MTBD was fit into the corresponding density using 'Jiggle-fit with Fourier filtering' option in *Coot*. The unstructured ends of MAP7 MTBD (residues 59 to 62 and 152 to 170) were removed based on DSSP (**D**efining the **S**econdary **S**tructure of **P**roteins) assignment in ChimeraX[71]. Single copies of each tubulin monomer were duplicated and fit on either side of the central tubulin heterodimer and all five chains were combined into one model. The geometry of the model was optimised using interactive molecular dynamics force field-based model fitting in ISOLDE[72], manual checking in *Coot* and real-space refinement in PHENIX. Before the last round of real space refinement, the chain corresponding to MAP7 MTBD was truncated to a poly-alanine backbone model in *Coot*. All images were made using Chimera[73,74] or ChimeraX[75].

### Preparation of solid-state NMR samples

For the preparation of solid-state NMR samples lyophilized tubulin (Cytoskeleton, Inc.) was solubilised in BRB80 buffer (80 mM PIPES, 2 mM MgCl₂, 1 mM EGTA, pH 6.8, 1 mM NaN₃, 1 mM DTT, pH 6.8 supplemented with protease inhibitor (Sigma-Aldrich, cOmplete EDTA-free), to a final concentration of 2 mg/mL). Then 1 mM Guanosine-5′-triphosphate (GTP) was added and incubation took place for 15 min at 30 °C. In the following, 20 μM paclitaxel (Taxol, SIGMA) was used to stabilize the MT and incubation took place for another 15 min at 30 °C. The MT were spun down at 180,000 g (Beckman TLA-55 rotor) for 30 min at 30 °C and the pellet was resuspended in warm BRB80 buffer with 20 μM paclitaxel. Subsequently 0.55 mg/mL [¹³C-¹⁵N]-MAP7 MTBD was added. The interaction partners were incubated for 30 min at 30 °C. In the following step [¹³C-¹⁵N]-MAP7 MTBD in complex with MT was separated from the unbound, non-polymerised fraction by centrifugation at 180.000 g (Beckman TLA-55 rotor) for 30 min at 30 °C. Afterwards, the pellet was washed with BRB80 buffer containing protease inhibitor, without disturbing the pellet. A 1.3 mm rotor was packed with the pellet.

### Solid-state NMR

Solid-state NMR experiments on [¹³C-¹⁵N]-MAP7 MTBD with Taxol-stabilized MT were measured on a 700 MHz Bruker Avance III spectrometer with a 1.3 mm MAS rotor, at 55 kHz with a set temperature of 260 K, resulting in an effective sample temperature of approximately 299 K. All experiments were carried out with a ¹H/X/Y triple-resonance MAS probe. Scalar-based 2D hCH were run and 3D hCCH correlation experiments were executed with 0 ms and 11 ms DIPSI[76,77] mixing time[78]. Dipolar-based sequences were used with cross-polarization (CP) steps with an amplitude ramp of 80-100 % on ¹H and 13 kHz PIS-SARO decoupling[79] during detection periods. CP transfer times were set to 700 μs forward CP and 150 μs back-CP. The dipolar 2D hCH and 3D hCCH experiments were recorded with 0 ms, 1.7 ms and 3.4 ms RFDR[48] mixing times. MISSISSIPPI was used for water suppression[80]. All spectra were processed with the Bruker TopSpin 3.6.2 software. The data was zero-filled and an EM window function with a LB of 120 Hz for the dipolar spectra and one of 30 Hz for the scalar ones was applied. Linear prediction in the indirect ¹³C dimensions was utilized with acquisition times (ms) of direct-/indirect-dimension of 24/32/32 for the 3D dipolar, 30/20 for the 2D dipolar, 32/12 for the 2D scalar and 32/18/18 for the 3D scalar. Chemical shifts were referenced via the water resonance. The 2D dipolar spectrum was measured with 64 scans and a spectral width of 18 × 80 ppm (direct-/indirect-dimension). For the 3D dipolar spectra, 32 scans were acquired, using a spectral width of 15 × 70 × 70 ppm (direct-/indirect-dimensions). The 2D scalar had 176 scans and a spectral width of 18 ×80 ppm (direct-/indirect-dimension). For the 3D scalar 24 scans were acquired and a spectral width of 18 × 80 × 80 ppm used (direct-/indirect-dimensions).

The spectra were analysed using POKY from NMRFAM-Sparky[81]. The residue abundance was estimated by signal integration of the peak position/region of interest and subsequent normalization to the integrated signal intensity average of well-resolved individual peaks that must represent a single resonance. For the scalar-based experiments, these resolved peaks refer to Ile 86 HG2CG2, Thr 146 CBHB, Asn 158 CBHB and Arg 159 HDCD resonances while to the dipolar data these correlations stem from Ser HBCB, Val CGHG, Leu CBHB and Ala CAHA. Resonance assignments for these reference peaks are highlighted in the Supplementary Table 2.

### Solution-state NMR

Solution-state NMR measurements for titration with MT were performed in NMR buffer (40 mM NaPi pH 6.5, 150 mM NaCl, 1 mM DTT) with 1 mM NaN₃ and 1 mM DTT, in 10 % D₂O. Two-dimensional (2D) ¹H-¹⁵N TROSY experiments of starting concentration 30 μM [¹³C-¹⁵N]-labelled MAP7 MTBD, were recorded at 295 K on a 600 MHz Bruker Avance III spectrometer equipped with a triple resonance cryogenic-probe. All spectra were processed using the Bruker TopSpin 3.6.2 software.

For the experiments with carboxy-terminal tails, peptides with the sequences SVEGEGEEEGEEY (α-CTT) and DATAEEEEDFGEEAEEEA (β-CTT) were purchased from Sigma-Aldrich and used without further purification. For the experiments the peptides were solubilised in NMR

buffer without NaPi and their pH adapted to pH 6.5 by adding Na$_2$HPO$_4$ and subsequently NaPi to obtain a concentration of 40 mM. For the NMR measurements increasing stoichiometric ratios of peptides to MAP7 MTBD were added to 80 μM [$^{13}$C-$^{15}$N]-MAP7 MTBD in NMR buffer. The 2D $^1$H-$^{15}$N TROSY were recorded at 298 K on a 900 MHz Bruker Avance III spectrometer.

## Spectral analysis of 2D and 3D ssNMR data sets

The analysis of scalar and dipolar 2D $^{13}$C-$^1$H correlation experiments in Fig. 3a was performed on the basis of previous solution NMR assignments and average BMRB chemical-shift values. For example, in panel (I), we find good agreement between peak positions for the Ala, Val and Leu side-chain resonances seen in the complex using J-based ssNMR (red) and solution-state results (crosses). In the same vein, correlations corresponding to Met 147 CE-HE, Lys 154 CG-HG and two peaks corresponding to Ile 86 QG1-HG1 and Ile 86 CD1-QD1 (where Q stands for one peak for multiple protons) are visible. On the other hand, most Lysine CG-QG side-chain correlations seem to appear in a spectral region only detected in the dipolar spectrum. Likewise, His and Trp CH-HB resonances seem to only appear in panel (II) in the dipolar-based ssNMR data. Region (III) shows side-chain resonances of Glu, Arg, Gln, Lys, Val, Leu, and Pro. In this region, both dipolar and scalar signals can be seen. Interestingly, Ile 86 CB-HB correlations seen in solution appears rigid (Blue, indicate in panel III), while as mentioned above for panel (I), Ile CD-QD and CG-QG might be flexible as there is only one Ile residue in the sequence. Region (IV) contains Arg CD-QD, Asp CB-HB, Tyr 99 CB-HB and Asn 158 CB-HB resonances. With some exceptions, MAP7 MTBD seems to have rigid Arg CD-QDs, suggesting stabilization of arginine sidechains after complex formation. On the other hand, the solution-state NMR CB-HB resonances associated with Asp 64, Asp 65 and Asn 158 are in good agreement with the scalar signals in the complex. The Leu CB-HB region that is depicted in panel (V) gives scalar, as well as dipolar signals. Panel (VI) shows the crowded CA-HA region. There are no resonances found for Pro CD-QD. Flexible CA-HAs include Ala CA-HA, several Glu CA-HAs and CA-HA of Asp 64, Asp 65, Asn 158 and His 167. For the crowded Arg, Glu, Leu, Gln, and Lys region both dipolar and scalar signals appear. Lastly, panel (VII) depicts well-dispersed Pro, Ser, Val, Ile CA-HA resonances that only appear in the mobile ssNMR experiments and correlate well with the solution assignments of free MAP7. Particularly, Pro 59, Val 60, Ile 86, Val 87, Pro 153, and Pro 170 overlap. In addition, it is intriguing to note that the Gln CG-HG region is mainly visible in the dipolar spectrum.

## 3D ssNMR data analysis

For the analysis of the 3D scalar spectrum (probing dynamic MAP7 residues in the complex) we observed good agreement with the solution-state assignments. Therefore, we were able to assign several residues of MAP7 MTBD in the MT-bound state (Fig. 4a, c). For Pro 59, Val 60, Leu 61, Val 63, Asp64, Leu 69, Glu 148, Pro 153, Gln 155, Asn 158, Arg 159 and Pro 170, resonances for both backbone and sidechains could be identified (Figs. 4a, 5a) while for Ala 70 and Ala 82 only CB resonances were found. In addition, we could tentatively assign Ile 86 CG and CD, Thr 146 CA-CB, CG, Met 147 CA-CE, CE; for Lys 154 CD-CG and Leu 166 CD-CG. Intriguingly, all of these residues are located at the C- and N-terminus of MAP7 MTBD (Fig. 5a), in line with weak or no binding to MT for MAP7 residues Pro 59-Ala 70 and Glu 148-Pro 170. In addition, several resonances corresponding to residue types could be partly identified in the scalar CCH 3D. Namely, one Glu, Ser, Gln, Leu and two additional Arg (Fig. 4a). Comparing these residues with the MAP7 MTBD sequence showed that they are present at the extreme N- or C-terminus. Furthermore, we compared the overall abundance of non-overlapping resonance types to the number of resonances of that type present in the aforementioned terminal domains, suggesting that the experimental signal intensity observed in our scalar based experiments is in reasonable agreement with the relative occurrence of

amino-acids in the protein segments Pro 59-Ala 70 and Glu 148-Pro 170 (Fig. 5b, Supplementary Fig. 8b). The residue abundance was calculated by dividing the integrated intensities of a resonance region by the average of integrated signal of separated peaks corresponding to one residue. For the scalar experiments, this average was taken from non-overlapping Leu HDCD, Leu HBCB, Arg HDCD and Ile HG2CG2 resonances, and for the dipolar experiment, the average was taken from non-overlapping Ser HBCB, Val CGHG, Leu CBHB and Ala CAHA resonances. Note that our spectral analysis estimates a larger fraction of flexible Leu side chains than present in the N- and C- terminal region, which would be compatible with sidechain motion even in the helical binding region. For example, Leu 103 is located between tubulins and might therefore exhibit flexibility. For the comparison of helicity of solution and solid-state MAP7 assignments (Fig. 5d), we used published NMR chemical-shift statistics[82] showing that α-helical chemical shifts (in ppm) are in general larger (CA) or smaller (CB) than the corresponding random-coil values. Hence, we calculated the difference Δ between the chemical-shift observed in the dipolar ssNMR spectra CS(CA/CB (ssNMR)) and the chemical shift obtained in the solution NMR spectra CS(CA/CB (solutionNMR)) in ppm. Resulting positive values for Δ in Fig. 5d for Δ(CA) or negative values Δ (CB) hence reveal an increased α-helical propensity for this residue type in our ssNMR data.

## Analysis of putative MAP7 MTBD A83-V87 beta-strand

By verifying connections in the CAHA region of the dipolar spectrum corresponding to the aforementioned β-strand resonances, we could identify Ala, Glu, Arg and a CAHA correlation corresponding to Val, Ile or Thr (Supplementary Fig. 8b). The presence of an Ala is supported by the peak integral of the residue in the CCH spectra which exceeds the expected number of Ala in the MAP7 sequence. (Fig. 5b, c). The same is the case for Val. Therefore, the CAHA correlation attributed above to Val, Ile or Thr, most likely reflects a Val residue (Supplementary Fig. 8c, right). Additionally, the resonance could be attributed to Ile 86 because there are two resonances observed for Ile 86 CB-HB and CG-HG in the dipolar 2D spectra, even though the MAP7 MTBD sequence contains only one Ile (Fig. 4b). Hence, this correlation remains unidentified. The residues that we identified for the β-strand resonances agree with residues Ala 83, Arg 84, Glu 85, Ile 86 and Val 87. These residues would be in line with the aggregation propensity for MAP7 evaluated by AGGRESCAN, which predicts residues 84–89 to be aggregation-prone[49]. Interestingly this region corresponds to the hinge, that was observed in the free MAP7 MTBD α-helix (residue 84–87)[17]. The rest of the aggregate could not be observed in the NMR spectra. This might be due to great heterogeneity or intermediate exchange dynamics. Finally, we estimate a relative ratio of 2.5: 1 between well folded (α-helical) MAP7 MTBD and aggregated MAP7 based on the signal integration in the corresponding CAHA regions (α-helical vs. *β*-strand) regions.

## Fluorescence anisotropy

0.1 mg α-CTT was dissolved in 100 μL NMR buffer (40 mM NaPi pH 6.5, 150 mM NaCl, 1 mM DTT) and the pH adjusted to 8.3 by adding 30 μL of 0.5 M Na$_2$HPO$_4$.The concentration of the α-CTT was measured with a BCA assay (as previously mentioned). 40 mM NaPi, 150 mM NaCl, pH 8.3 was added to obtain a 500 μM α-CTT concentration. From here, the experiment was performed in the dark due to the light-sensitivity of NHS-fluorescein. 4 times molar ratio NHS-fluorescein to α-CTT was incubated for 2 h at room temperature, inverted, whilst protected from light. The NHS-fluorescein-labelled α-CTTs was dialysed against NMR buffer (40 mM NaPi pH 6.5, 150 mM NaCl, 1 mM DTT) overnight at 4 °C in a 1 kDa cut-off dialysis tube, with a buffer exchange after 2 h, to remove unbound NHS- fluorescein. By measuring the absorbance at 494 nm, combined with the extinction coefficient of 68/M*cm and the concentration of α-CTT from the BCA the concentration of labelled α-

CTT was calculated. A dilution series of MAP7 MTBD resulting in a concentration range from 200 to 0 μM in a 384-well plate was prepared and NHS-fluorescein-labelled α-CTT to concentration of 1 μM was added to each well. We measured the fluorescence anisotropy at 494 nm with a plate reader and analysed the results with the DynaFit software (BioKin Ltd.)[22].

## Reporting summary

Further information on research design is available in the Nature Portfolio Reporting Summary linked to this article.

## Data availability

The cryo-EM map for MAP7 MTBD-stabilized MTs generated in this study have been deposited in the Electron Microscopy Data Bank under accession code EMD-19042 and the corresponding atomic model for four tubulin monomers and MAP7 MTBD helical backbone have been deposited in the Worldwide Protein Data Bank under accession code 8RC1. The cryo-EM maps for 13- and 14-protofilament Taxol-stabilized MAP7 MTBD-bound MTs generated in this study have been deposited in the Electron Microscopy Data Bank under accession codes EMD-19043 and EMD-19044. Corresponding C1 (for all protofilament maps) and symmetrized reconstructions (only for MAP7 MTBD stabilized MTs) have been deposited as additional maps under the respective Electron Microscopy Data Bank codes. NMR resonance assignments of MAP7 have been deposited in the Biological Magnetic Resonance Data Bank (BMRB) under the accession code [doi:10.13018/BMR51730]. All other data supporting the findings of this study are available in the article, the Supplementary Information and the Source data file or can be obtained from the corresponding authors upon request. Source data are provided with this paper.

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

## Acknowledgements

This work was supported by NWO (the Dutch Science Foundation) via a TOP-PUNT (grant number 718.015.001) grant to M.B., by uNMR-NL, the National Roadmap Large-Scale NMR Facility of the Netherlands (grant number 184.032.207 to M.B.),  the uNMR-NL grid (grant number 184.035.002 to M.B.) and by Medical Research Council, U.K. (MR/R00352/1) to C.A.M. We acknowledge Diamond Light Source for access and support of the cryo-EM facilities at the UK's national Electron Bio-imaging Centre (eBIC) funder proposal EM20287-56, funded by the Wellcome Trust, MRC and BBSRC. Cryo-EM data for the final EM reconstructions were collected at ISMB EM facility (Birkbeck College, University of London) with financial support from the Wellcome Trust (202679/Z/16/Z and 206166/Z/17/Z). We thank N. Lukoyanova and S. Chen for electron microscope support and D. Houldershaw for computational support at Birkbeck. M.B. and C.A.M. acknowledge J. Atherton's initial contributions in electron microscopy studies of MAP7 MTBD with microtubules. A.A. and M.B. would like to thank Anna Akhmanova for providing the MAP7 construct and for scientific discussions. A.A. thanks Dr. U. B. le Paige for fruitful discussions and ideas.

## Author contributions

A.A. carried out protein purification, MT assemblies, the solid-state NMR measurements with the help of S.B. Solution-state NMR experiments were conducted by A.A. with supervision of H.v.I. J.W.B. conducted the SEC-MALS experiments. A.A. and M.B. wrote the paper with input from all authors. M. Ban conducted the EM experiments. M. B. and C.A. M. supervised the project.

## Competing interests

The authors declare no competing interests.
