## [Peer Review File · Nature Communications]

A structural and dynamic visualization of the interaction between MAP7 and microtubulesREVIEWER COMMENTS

Reviewer #1 (Remarks to the Author):

This is an excellent study into the structural basis of interactions of microtubule-binding domain (MTBD) of MAP7 protein with polymerized microtubules. MAP7 both binds to microtubules and activates kinesin-1-mediated cargo transport along microtubules by an unknown mechanism. The authors use a combination of solid-state NMR, electron microscopy, fluorescence anisotropy, and isothermal titration calorimetry to shed light onto the structural determinants of MAP7's 112-residue large MTBD interactions with microtubules. The main conclusions of this study are that: i) MTBD binds along the microtubule protofilament and stabilizes the microtubule lattice; ii) MTBD forms an extended alpha-helix and retains dynamic N- and C-termini upon binding with microtubules; iii) MTBD engages in interactions with microtubules' CTT through two regions. The latter finding is very important as it is a direct demonstration of the role of CTT's in recruiting MAP7 (and other MAPs) through predominantly electrostatic interactions.

Overall, the technical quality of the work is superb, the study establishes a structural model of MAP7-microtubule interactions with atomic level detail, and is an important milestone in understanding the mechanism of MAP7 mediated kinesin-1 recruitment to microtubules. The work will be of interest to the broad readership of Nature Communications, and I enthusiastically recommend its publication.

A couple of minor typos:

p. 11, line 276 – “due to static conformational heterogeneity”

p. 13, line 308 – “micromolecular” should be “micromolar”?

Reviewer #2 (Remarks to the Author):

The authors present high-quality experimental data that may provide a further insight into MAP7-MT interaction. However, it is not clearly described what is new and what experimental data reproduce previously obtained results. In particular, it should be described if the presented cryo-EM data provide new information or if they are supposed to confirm results recently published by Ferro et al. (properly cited in the manuscript as Ref. 18). It seems that starting with the NMR results would result in a better flow of the arguments.

The conclusions are supported by experimental data, the methodology is sound and well chosen, the data analysis is correct. Presentation of the results should be improved and some additional details provided as suggested below.

RESULTS:

Section "MAP7 MTBD binds with micromolar dissociation constant to MTs"

It is not clear if the reported error of the ITC measurement is a standard deviation of multiple measurements (at least triplicate measurement is desirable) or a standard deviation of the data from the fitted curve (as suggested by presenting a single plot). This should be clarified.

Section "MAP7 MTBD binds along the MT protofilament and stabilizes the MT lattice"

Details of model building are not well described. How was the direction of the helix and its register actually determined? Based on sufficiently resolved electron density of some side chains? By comparison with previously published structures (Ferro et al. 2022)?

The authors compare their cryo-EM data with the AlphaFold2 prediction. Is it the best model to be presented? The C-terminal region of the MTBD helix deviates from the electron density in Fig. 2b. It should be commented. Superposition with the structural model in Fig. 2c would also help (Fig. 2b is too small). Comparison with the NMR data (described earlier in the paper) would be useful.

The authors mention that their structure matches the results obtained with Taxol-stabilized microtubules in Ref. 18 (Ferro et al. 2022), but do not clarify whether they talk about full-size MAP7 or about the MTBD fragment. Also, a match with the electron density, not only with the helix cartoon, could be presented in Fig. S3d.

As the NMR measurements are performed with Taxol, one would expect that the authors acquire cryo-EM data also on their sample stabilized with Taxol to gain a direct comparison.

Section "MAP7 MTBD shows an extended alpha-helix and dynamic terminal regions upon MT binding"

The first reference to Fig.3a when introducing the solid-state NMR experiments is misleading as Fig.3a shows the assignment in solution.

It is not clear what is the relation of chemical shifts presented in Supplementary Table 1 to those in BMRB deposition, accession number 51,730 (some numbers are identical, but some shifts are missing). Supplementary Tables 1 and 2 are not described sufficiently in their captions.

New assigned chemical shifts should be deposited (BioMagResBank).

The caption of Fig.3 should say signals of what atoms attributed to the beta-sheet are presented in the purple dashed box.

The alteration of the rigid/flexible residues seems to roughly match the periodicity of the alpha helix. Could the authors directly say that the rigid residues are oriented towards MT and flexible residues point outside (rather than talking about amino acid types)?

When talking about identification of helical residues based on chemical shifts, a graphical representation should be shown (like the TALOS-N prediction as in Ref. 17).

It is inconsistent (and somewhat confusing) to label assigned and unassigned residues in the same manner in Fig. 4 and Supplementary Table 2 (e.g. A3 vs D84). A, A2, A3 should be changed e.g. to Aa, Ab, Ac etc.

Section "Two protein regions play a role in the MT carboxy-terminal tail interaction with MAP7 MTBD"

Two typos: "micromolecular" should read "micromolar", "reduction of chemical shifts" should read "reduction of chemical shift perturbations".

DISCUSSION

A direct comparison with the highly relevant Ref. 18 (Ferro et al.) should be presented in Discussion.

Reviewer #3 (Remarks to the Author):

In this paper, Adler et al. employ a range of techniques, and in particular solid and solution state NMR in combination with cryo-EM, to investigate the structure and interaction of MAP7 (microtubule-associated protein 7) with microtubules. The use of NMR as exemplified in this work is a promising approach to investigate the structure and dynamics of microtubule binding proteins. However, the quality of the cryo-EM work is not commensurate with the claims of the paper and the reconstruction appears of less resolution than one recently reported by another group of the same complex. The primary novel information derived from the NMR data is that the region of interaction between the MAP7 MTBD and the microtubule extends beyond what can be visualized by previous cryo-EM work and that the C- and N-terminal regions of the MTBD are mostly dynamic and may form low-affinity charged interactions with the tubulin C-terminal tails. These results increase our understanding of the structure of the MTBD-microtubule complex but offer limited novel insights into the physiological role of MAP7 or its mechanism of action. Thus, overall, the work appears minimally incremental.

Specific comments:

1) The abstract states as one of the main conclusions of the paper: "Our results show that a combination of interactions between MAP7 and MT lattice extending beyond a single tubulin dimer"

This is not a novel finding. This was already shown by Ferro et al (ref 18) where they determine by cryo-EM the microtubule binding site of the MAP7 MTBD and that the MTBD forms an α -helix that extends beyond a single tubulin dimer and interacts with the next tubulin subunit on the (-) end position.

2) In page 3. the authors say: "Our results suggest an interaction with micromolar affinity between an extended α -helix and two tubulin dimers; this expands the previously determined MAP7-MT binding interface by around 25 residues"

Some clarification is needed here. Fig. 5A shows the extent of the MTBD derived from previous cryo-EM data and the NMR data. It indicates that according to the NMR data the MTBD α -helix extends from the helix visualized by cryo-EM. What is the evidence that all these additional α -helix residues interact directly with the microtubule? In particular, the interesting detection of some prevalent beta-sheet indicate that there are at least 2 forms of MAP7 MTBD observed: the fully helical version the authors focus on and another version where a proposed stretch of residues (83 to 87) form beta-sheet which the authors interpret as aggregation of MAP7. This raises the question whether some of the helix signal observed in the solid state NMR spectra could also arise from these aggregates and not only from microtubule bound MAP7. An estimate of the apparently large amount of MAP7 present as aggregates is not provided. The presence of multiple MAP7 conformations or interacting modes also raises the question whether all the MAP7-microtubule proposed interactions can occur on the same MAP7 molecule.

Note also that the limits of the newly NMR identified areas indicated in Fig 5A include 26 not 25 residues and that the previously reported MTBD structure in complex with the microtubule (PDBid: 7SGS) go from W87 to H139., not 83 to 134 as indicated in Fig. 5A. Please clarify.

3) Page 5: "This, together with the a priori probability of MAP7 MTBD interacting out of phase with tubulin dimers along and between protofilaments around the MTs, resulted in lower resolution of the MAP7 MTBD density ($\sim 4 \text{ \AA}$) compared to the rest of the structure,"

The density shown in Figure 2C suggests that 4 \AA is an overoptimistic estimate of the resolution in this area. The 3.5 \AA FSC0143 estimate most likely applies to the microtubule area which constitutes the larger volume with the strongest densities in the map. If the MTBD was resolved at $\sim 4 \text{ \AA}$ then the helical path of the MTBD α -helix and some side chains should be resolved which does not appear to be the case. A close-up view of the MTBD with the fitted model (with side chains) should be shown to support a claim of $\sim 4 \text{ \AA}$ resolution in this area. It is also unfortunate that the structure of the MTBD beyond a single tubulin dimer could not be visualized. The cryo-EM data confirms the azimuthal position of the MTBD around the microtubule as determined by Ferro et al., but because neither the C- or N terminal boundaries or key residues could be identified it cannot confirm, refute or refine the axial register of the MTBD on the microtubule estimated by Ferro et al. Therefore the following statement by the authors regarding the registration is not proven experimentally: "The binding site and register of the MAP7 MTBD helix on the MTs also matches that obtained from 3D reconstruction of MAP7 MTBD bound to Taxol stabilized-MTs18, demonstrating that MAP7 interacts with MTs in the same way independently of how the MTs are stabilized (Supplementary Fig. 3d)".

As it is, the cryo-EM data presented in this paper does not add information to the cryo-EM microtubule-MAP7-MTBD complex structure recently reported by Ferro et al. (Science 2022, ref 18). The resolution of the MTBD in the Ferro et al. reconstruction with resolved side chains also appears better than in this work.

4) When the authors refer to the symmetrized reconstruction it is not fully clear what it is the extent of the symmetrization. Are all tubulin dimers and bound MTBD averaged or all tubulin monomers (alpha and beta) with bound MTBD? The second case would occur if the presence of seams in the microtubule images were disregarded or if their location in all microtubule images could not be accurately determined. If the claim is the former as suggested by the EM methods paragraph, then some evidence should be presented showing that indeed the alpha and beta tubulin structures are separated and that they show distinct structural features corresponding to their slightly different amino acid sequences. Note that averaging all monomers either by design or by accident may partially account for the poor resolution of the MTBD in the reconstruction. In any case the averaging procedure used was not adequate given that the proposed MTBD model (Fig. 7) extends almost completely over two tubulin dimers.

5) The fitting of the AlphaFold model to the cryo-EM structure is poor. In Fig 2b the upper part of the MTBD model is completely out of the cryo-EM density. This is under-emphasized, and the authors do not show a zoom-in view of their AlphaFold MAP7 model in their cryo-EM density (the MAP7 model is not shown in Fig. 2c for instance). A closer view of the alpha fold model fitted in Fig. 2b is provided in Supplementary Fig. 3.d but without a cryo-EM density overlay. Additionally, the alpha fold model also deviates substantially from the experimental model from Ferro et al in the part of MAP7 nearer tubulin alpha-3 (Supplementary Fig 3d). The AlphaFold model could have been useful to support the fact that the new interaction the authors propose would be possible structurally, however it appears it is not the case from the above considerations.

6) The authors indicate on page 15: "The register is supported by the observation of more flexible residues in the clefts between the tubulin subunits" However this is only partially true. For example, according to the Ferro et al. model MTBD residue R106 is at the middle of the gap between tubulin subunits yet this residue and the adjacent ones are shown to be 'rigid'. Overall, the otherwise interesting differences of dynamics obtained from the NMR experiments are not sufficiently clear to assign a precise registration of MAP7 and unfortunately this information cannot be derived from the cryo-EM data presented in this paper either.

7) In the summary structural model shown in bottom left panel of Figure 7 where two tubulin dimers are shown, some residues of the MTBD should be indicated to better relate it to areas mentioned in the text and other figures and to compare with other models. For example, does the segment between MTBD residues 86-139 is placed on the microtubule as in the Ferro et al. model? If yes, this should be acknowledged. The Figure title says 'based on the cryo-EM' but does not specify which cryo-EM data.

8) Regarding the solution-state NMR experiment with the tubulin tails mimics, the authors mention "For the experiments the peptides were solubilized in NMR buffer and their pH adapted." The last statement about the pH is somewhat ambiguous: was the final buffer and pH of the 2 solutions used in the titration the same?

9) The authors indicate on Page 5: "Further, extension of density for both α - and β -tubulin's H12 helix could be observed in the reconstruction and additional C-terminal residues Val437-Ser439 for α -tubulin and residues Ala428-Thr429 for β -tubulin were modelled into the density (Table 2). The ordering of

these usually disordered residues could be explained by an interaction between the tubulin CTTs and MAP7; however, the direct interaction is not visible in our reconstruction and is presumably less ordered (Fig. 2c)."

This statement on the stabilization of H12 and C-terminal residues is not supported by the data provided in the paper. One would need to compare the densities of the residues mentioned here with the ones present in a naked microtubule obtained at a similar resolution.

10) "the corresponding atomic model for the tubulin dimer is available in the PDB with accession number 8P6R."

On a tubulin dimer only? The full model of the MTBD bound to two tubulin dimers (i.e., the model shown in Fig. 7) is what it needs to be deposited.

Point by Point response: Adler, Bangera et al. A structural and dynamic visualization 1 of the interaction between the microtubule-associated protein 7 (MAP7) and microtubules, NCOMMS-23-23984

Reviewer #1 (Remarks to the Author):

[1] This is an excellent study into the structural basis of interactions of microtubule-binding domain (MTBD) of MAP7 protein with polymerized microtubules. MAP7 both binds to microtubules and activates kinesin-1-mediated cargo transport along microtubules by an unknown mechanism. The authors use a combination of solid-state NMR, electron microscopy, fluorescence anisotropy, and isothermal titration calorimetry to shed light onto the structural determinants of MAP7's 112-residue large MTBD interactions with microtubules. The main conclusions of this study are that: i) MTBD binds along the microtubule protofilament and stabilizes the microtubule lattice; ii) MTBD forms an extended alpha-helix and retains dynamic N- and C-termini upon binding with microtubules; iii) MTBD engages in interactions with microtubules' CTT through two regions. The latter finding is very important as it is a direct demonstration of the role of CTT's in recruiting MAP7 (and other MAPs) through predominantly electrostatic interactions.

Overall, the technical quality of the work is superb, the study establishes a structural model of MAP7-microtubule interactions with atomic level detail, and is an important milestone in understanding the mechanism of MAP7 mediated kinesin-1 recruitment to microtubules. The work will be of interest to the broad readership of Nature Communications, and I enthusiastically recommend its publication.

Response 1): We thank the reviewer for the positive comments on our work.

[2] A couple of minor typos:

p. 11, line 276 – “due to static conformational heterogeneity”

Response 2): We thank the reviewer for pointing this out to us. These typos have been corrected in the revised version (line 310).

[3] p. 13, line 308 – “micromolecular” should be “micromolar”?

Response 3): We thank the reviewer for this comment which has been addressed in the revised version (line 342).

Reviewer #2 (Remarks to the Author):

[1] The authors present high-quality experimental data that may provide a further insight into MAP7-MT interaction. However, it is not clearly described what is new and what experimental data reproduce previously obtained results. In particular, it should be described if the presented cryo-EM data provide new information or if they are supposed to confirm results recently published by Ferro et al. (properly cited in the manuscript as Ref. 18). It seems that

starting with the NMR results would result in a better flow of the arguments.

Response 4): Our originally presented cryo-EM data provided information about the MAP7-MTBD (residues 59-170) binding site on MAP7-MTBD-nucleated and stabilized MTs in the absence of bound drug. This is in contrast to the work by Ferro et al, where pre-stabilized drug-bound MTs were used to determine the structures of several MAP7 constructs (full-length MAP7, residues 60-170, residues 83-134). In these experiments, MTs were stabilized by the drug peloruside, which was previously shown by Kellogg et al (PMID: 28104363, ref. 53 in our paper) to bind on the outer surface of MT lateral contacts, ~15 Å from the MAP7 binding site. The previous version of our manuscript incorrectly stated that the MTs were stabilized by Taxol in this study, and we apologize for the confusion arising from that error. We have now included a new cryo-EM reconstruction of MAP7-MTBD (residues 59-170) bound to Taxol-stabilized MTs (Supplementary Fig. 6). This shows the same binding site as we originally observed in the absence of drug, which is also where Ferro et al. (PMID: 35050657) observed MAP7 binding. All the cryo-EM structures reported to date are completely consistent about the MAP7 binding site on MTs and, because the structures are determined from MTs stabilized in different ways, collectively indicate that MAP7 MT binding is not influenced by the mechanism by which MTs are nucleated and stabilized, at least in vitro. This is an important new finding and also reinforces the relevance of the NMR data collected using Taxol-stabilized MTs. We have elaborated further on these comparisons in the text on page 6 and page 16 of our revised manuscript.

In the context of the additional data we have added, and further data requested by the reviewers, we have substantially rewritten the manuscript, but have concluded that presenting the EM data first links more clearly to prior knowledge, its limitations due to averaging, and from which unique insights from NMR experiments logically develop. We therefore feel that the original order of data presentation is overall more coherent for the reader, but would be happy to follow editorial guidance on this point.

[2] The conclusions are supported by experimental data, the methodology is sound and well chosen, the data analysis is correct.

Response 5): We thank reviewer for these comments.

[3] Presentation of the results should be improved and some additional details provided as suggested below. RESULTS:

Section "MAP7 MTBD binds with micromolar dissociation constant to MTs"

[4] It is not clear if the reported error of the ITC measurement is a standard deviation of multiple measurements (at least triplicate measurement is desirable) or a standard deviation of the data from the fitted curve (as suggested by presenting a single plot). This should be clarified.

Response 6): We have clarified in the caption of Figure 1 that the experiment was carried out in duplicate and that representative ITC data are shown. The raw data are included in our revision submission the in data source file.

[5]Section "MAP7 MTBD binds along the MT protofilament and stabilizes the MT lattice"

Details of model building are not well described. How was the direction of the helix and its register actually determined? Based on sufficiently resolved electron density of some side chains? By comparison with previously published structures (Ferro et al. 2022)?

The authors compare their cryo-EM data with the AlphaFold2 prediction. Is it the best model to be presented? The C-terminal region of the MTBD helix deviates from the electron density in Fig. 2b. It should be commented. Superposition with the structural model in Fig. 2c would also help (Fig. 2b is too small). Comparison with the NMR data (described earlier in the paper) would be useful.

The authors mention that their structure matches the results obtained with Taxol-stabilized microtubules in Ref. 18 (Ferro et al. 2022), but do not clarify whether they talk about full-size MAP7 or about the MTBD fragment. Also, a match with the electron density, not only with the helix cartoon, could be presented in Fig. S3d.

Response 7): The reviewer raises very reasonable points about the lack of clarity of presentation of our previous data and methods. In brief, we previously used the AlphaFold2-derived model approximately docked within the EM density to visualize this interaction. We also compared the similarity of these coordinates with those modelled by Ferro et al (former Supplementary Fig. 2d). However, as the reviewer notes, there were a number of limitations to this approach, including the protrusion of part of the AlphaFold2 model from the EM density. This is because prior information used by AlphaFold model calculation is dominated by curved tubulin X-ray crystal structures, which in turn caused the original MAP7 model to curve alongside. Retrospectively, we apologize this was not clear and we have completely redone this part of our analysis. We have added more data and rewritten much of the relevant text (Methods, Results and Discussion). We have also added further Figure panels (Supplementary Fig. 4d, 5 and 6) to provide a more thorough cross-comparison with previous work in the resubmitted version.

[6] As the NMR measurements are performed with Taxol, one would expect that the authors acquire cryo-EM data also on their sample stabilized with Taxol to gain a direct comparison.

Response: 8) As noted above, we have now included a reconstruction of the identical MAP7-MTBD construct bound to Taxol-stabilized MTs, in addition to the originally presented cryo-EM reconstructions (now also with more data) in which MTs were directly stabilized by the presence of MAP7. We have included additional Results, Figure panels (Supplementary Fig. 6) and Discussion to describe our findings from this comparison and more robustly support cross-comparison with the NMR data.

[7] Section "MAP7 MTBD shows an extended alpha-helix and dynamic terminal regions upon MT binding"

The first reference to Fig.3a when introducing the solid-state NMR experiments is misleading as Fig.3a shows the assignment in solution. It is not clear what is the relation of chemical shifts presented in Supplementary Table 1 to those in BMRB deposition, accession number 51,730

(some numbers are identical, but some shifts are missing). Supplementary Tables 1 and 2 are not described sufficiently in their captions. New assigned chemical shifts should be deposited (BioMagResBank).

Response 9): We thank the reviewer for this comment. For the sake of clarity, we have modified text and Figure 3 which now starts with the ssNMR CH spectrum in Fig. 3a. In Figure 3b, the solution-state NMR data are shown and we also modified the color coding in Figure 3b to improve readability. Notably, only CH correlations are shown and we did not utilize NH shifts obtained in ref. 17 and deposited in the BMRB. Furthermore, additional information is given in the captions of SI tables 1 and 2. As recommended by the reviewer we have been in contact with the BioMagResBank and our BMRB submission has been updated to include the additional assignments mentioned in our original submission.

[8] The caption of Fig.3 should say signals of what atoms attributed to the beta-sheet are presented in the purple dashed box.

Response 10): We have provided this information in the revised caption of Figure 3 and also made reference to it in supplementary Figure 7c.

The alteration of the rigid/flexible residues seems to roughly match the periodicity of the alpha helix. Could the authors directly say that the rigid residues are oriented towards MT and flexible residues point outside (rather than talking about amino acid types)?

Response 11): We thank the reviewer for this very valuable comment. Interestingly, we observed several flexible residues in the α -helical region with a spacing of 12, 13 (12 +13 = 7 helix turns), 27 (7.5 helix turns), 19 (5.3 helix turns), 6 (1.7 turns) and 6 (1.7 turns) when counted from the first flexible residue after a rigid sequence until the subsequent flexible residue. This could indeed reflect residues pointing outward from the MAP7 MTBD- MT binding interface. However, we feel that further residue-specific assignments should be obtained to substantiate such conclusions. Because of the unfavorable amino-acid sequence of MAP7, such NMR experiments would require tailored (possibly amino-acid specific) isotope labelling which is outside the scope of our current study.

[9] When talking about identification of helical residues based on chemical shifts, a graphical representation should be shown (like the TALOS-N prediction as in Ref. 17).

Response 12): We thank the reviewer for this suggestion. Unlike our analysis in ref. 17, a residue-specific analysis of our dipolar data was not possible. For this reason, we resorted in Figure 5d to a more simplified analysis in which we used published NMR chemical-shift statistics (PMID: 11910028) that show that α -helical chemical shifts (in ppm) are in general larger (CA) or smaller (CB) than the random-coil values.

We added a corresponding statement in the SI (lines 164-170) to explain our procedure. Also, we included “[ppm]” to the vertical axis of Figure 5d and in the caption to further clarify our approach.

[10] It is inconsistent (and somewhat confusing) to label assigned and unassigned residues in the same manner in Fig. 4 and Supplementary Table 2 (e.g. A3 vs D84). A, A2, A3 should be changed e.g. to Aa, Ab, Ac etc.

Response 13): We thank the reviewer for this suggestion. We have modified supplementary table 2 accordingly.

[11] Section "Two protein regions play a role in the MT carboxy-terminal tail interaction with MAP7 MTBD"

Two typos: "micromolecular" should read "micromolar", "reduction of chemical shifts" should read "reduction of chemical shift perturbations".

Response: 14): Thank you. These have been addressed in the revised version in lines 342 and 345, respectively.

[12] DISCUSSION

A direct comparison with the highly relevant Ref. 18 (Ferro et al.) should be presented in Discussion

Response 15): This is an important point, especially given the additional data that we have now included, and we have included this comparison in the Discussion (p16) lines 390-413 and Supplementary Fig. 6.

Reviewer #3 (Remarks to the Author):

[1] In this paper, Adler et al. employ a range of techniques, and in particular solid and solution state NMR in combination with cryo-EM, to investigate the structure and interaction of MAP7 (microtubule-associated protein 7) with microtubules. The use of NMR as exemplified in this work is a promising approach to investigate the structure and dynamics of microtubule binding proteins. However, the quality of the cryo-EM work is not commensurate with the claims of the paper and the reconstruction appears of less resolution than one recently reported by another group of the same complex. The primary novel information derived from the NMR data is that the region of interaction between the MAP7 MTBD and the microtubule extends beyond what can be visualized by previous cryo-EM work and that the C- and N-terminal regions of the MTBD are mostly dynamic and may form low-affinity charged interactions with the tubulin C-terminal tails. These results increase our understanding of the structure of the MTBD-microtubule complex but offer limited novel insights into the physiological role of MAP7 or its mechanism of action. Thus, overall, the work appears minimally incremental.

Response 16: As is discussed in detail in response to specific points below and elsewhere in this rebuttal (e.g. Response 4), there were already several novel, experimentally-derived conclusions in our original submission and these have been further enhanced by additional data incorporated into the revised version of the manuscript. In overview, however:

i) Our original cryo-EM data showed how MAP7-MTBD binding along the MT protofilament can promote MT polymerization and stabilization (Supplementary Fig. 2) – this had not been shown before.

ii) We now also show that the binding site on MTs by which these activities are achieved is the same as observed when the same construct is bound to Taxol-stabilized MTs – this observation is also consistent with previously published work by Ferro et al, who used peloruside-stabilized MTs for their reconstructions. Our new data now robustly demonstrate that MAP7 MT binding is not perturbed by different modes of MT stabilization, which had also not been shown before (Supplementary Fig. 6).

iii) An important limitation of all these cryo-EM studies of MAP7 (both ours and that of Ferro et al) is that reconstruction procedures involve averaging across 80 Å tubulin dimers, which results in blurring of information about binding of the ~130 Å MTBD. Because of this, binding of MAP7 MTBD across more than one dimer can only be inferred indirectly from modelling experiments, using Rosetta in the case of Ferro et al and AlphaFold2 in our analysis. While this can be used to extract information about binding of the central fragment (87-134) of the MAP7 MTBD as in Ferro et al, structural analysis becomes more complicated when the entire MAP7 MTBD fragment is considered as the binding unit. Therefore, a critical new aspect of our study comes from the ssNMR analysis of the MAP7 MTBD bound to Taxol-stabilized MTs which provides direct experimental evidence of the full extent of the MTBD MT interaction. Although mentioned in the Ferro et al paper, our work now includes more extensive information about MAP7 MTBD than previously modelled and without blurring from averaging. Importantly, our described approach also provides a general framework for investigating dynamic, extended interactions between MTs and their binding partners.

The novelty of this specific insight from our current study has recently already been cited in a review by one of the teams involved in the Ferro et al work (PMID: 37702417). In that review, the authors cite the bioRxiv version of our current manuscript as evidence of binding across multiple tubulin dimers: “First visualized bound to MTs in the context of its interplay with kinesin 1 on the MT surface (Ferro et al., 2022), it turned up [sic] to have a footprint on the MT larger than a tubulin dimer repeat (Adler et al., 2023).”

iv) The involvement of tubulin CTTs in MT binding by MAP7-MTDB (Fig. 2c)

Apparently, the original version of our manuscript did not make these novel aspects of our study sufficiently clear. We have made substantial changes to the text and data presentation to address this issue.

Specific comments:

[2] 1) The abstract states as one of the main conclusions of the paper: "Our results show that a combination of interactions between MAP7 and MT lattice extending beyond a single tubulin dimer"

This is not a novel finding. This was already shown by Ferro et al (ref 18) where they determine by cryo-EM the microtubule binding site of the MAP7 MTBD and that the MTBD forms an α -

helix that extends beyond a single tubulin dimer and interacts with the next tubulin subunit on the (-) end position.

Response 17): While Ferro et al inferred the possibility of the entire MAP7 MTBD binding to the MT, their model and structural analysis is based on the 87-134 fragment being the conserved MT binding fragment. We use the complete MAP7 MTBD (59-170) in our AlphaFold 2 predictions and highlight the ambiguity in assigning the register of full MAP7 MTBD helix bound to the MT due to averaging over a tubulin dimer (Supplementary Fig. 4c). Further, as noted in Response 16), the presentation of direct experimental evidence of the MT-MAP7 MTBD interaction extending beyond a single tubulin dimer using NMR is a key novel aspect of our study. In the substantial rewrite that we have undertaken, we have sought to make the novelty of our findings clearer, and to explicitly discuss the comparison of our findings with Ferro et al, as also requested by Reviewer 2 (Response 15). We therefore think that this sentence in the abstract is indeed supported by the data and analysis we present.

[3] 2) In page 3. the authors say: "Our results suggest an interaction with micromolar affinity between an extended α -helix and two tubulin dimers; this expands the previously determined MAP7-MT binding interface by around 25 residues"

Some clarification is needed here. Fig. 5A shows the extent of the MTBD derived from previous cryo-EM data and the NMR data. It indicates that according to the NMR data the MTBD α -helix extends from the helix visualized by cryo-EM. What is the evidence that all these additional α -helix residues interact directly with the microtubule?

Response 18): The ssNMR samples are prepared after sample centrifugation, i.e., any unbound MAP7 will be removed from the ssNMR sample. In addition, the dipolar spectra only record rigid components at a sample temperature of 299 K. Hence, the ssNMR data should be dominated by MAP7 residues that are bound to the MTs. Consistent with that idea and as indicated in Figure 5d, those residues exhibit different Ca and Cb chemical shifts compared to our solution-state NMR data.

[4] In particular, the interesting detection of some prevalent beta-sheet indicate that there are at least 2 forms of MAP7 MTBD observed: the fully helical version the authors focus on and another version where a proposed stretch of residues (83 to 87) form beta-sheet which the authors interpret as aggregation of MAP7.

This raises the question whether some of the helix signal observed in the solid state NMR spectra could also arise from these aggregates and not only from microtubule bound MAP7.

Response 19): For reasons described in Response 18, and due to the fact that we clearly see a beta-sheet rich region that must stem from labelled MAP7, we find this highly unlikely. We also would like to point out that our findings of two different protein populations present in our ssNMR data is not new. For example, in our own previous work (PMID: 17199296) we simultaneously observed several protein conformations in an ssNMR sample.

[5] An estimate of the apparently large amount of MAP7 present as aggregates is not provided.

Response 20): We have added this information in the revised version of the SI (lines 188-191).

We also note that our interpretation of aggregate formation is further supported by sec-MALS data showing concentration dependent aggregation in solution. These data are now included in SI figure 7 d and e. Correspondingly, the caption (lines 90-95) has been extended and discusses the observation of aggregates. In addition, lines 193-200 of the SI discuss the details of the experiments and Wouter Beugelink who conducted these experiments is now a co-author on our manuscript.

[6] The presence of multiple MAP7 conformations or interacting modes also raises the question whether all the MAP7-microtubule proposed interactions can occur on the same MAP7 molecule.

Response 21): As mentioned under Response 19, we cannot exclude that some of MAP7 is not bound to MTs and gives rise to the aggregated protein signal in our ssNMR data sets. From our NMR data we cannot comment on where MAP7 binds. Using mixed labelled samples (e.g. using ¹³C labelled MT and ¹⁵N labelled MAP7) may help but such NMR experiments are challenging and outside the scope of our current study.

[7] Note also that the limits of the newly NMR identified areas indicated in Fig 5A include 26 not 25 residues and that the previously reported MTBD structure in complex with the microtubule (PDBid: 7SGS) go from W87 to H139., not 83 to 134 as indicated in Fig. 5A. Please clarify.

Response: 22) We thank the reviewer for this comment. Figure 5a has been modified accordingly. We also refer the reviewer to the updated Figure 7 and the discussion (lines 425-438) where the NMR findings are clarified.

[8] 3) Page 5: "This, together with the a priori probability of MAP7 MTBD interacting out of phase with tubulin dimers along and between protofilaments around the MTs, resulted in lower resolution of the MAP7 MTBD density (~4 Å) compared to the rest of the structure,"

The density shown in Figure 2C suggests that 4Å is an overoptimistic estimate of the resolution in this area. The 3.5 Å FSC0143 estimate most likely applies to the microtubule area which constitutes the larger volume with the strongest densities in the map. If the MTBD was resolved at ~4 Å then the helical path of the MTBD α-helix and some side chains should be resolved which does not appear to be the case. A close-up view of the MTBD with the fitted model (with side chains) should be shown to support a claim of ~4 Å resolution in this area. It is also unfortunate that the structure of the MTBD beyond a single tubulin dimer could not be visualized. The cryo-EM data confirms the azimuthal position of the MTBD around the microtubule as determined by Ferro et al., but because neither the C-or N terminal boundaries or key residues could be identified it cannot confirm, refute or refine the axial register of the MTBD on the microtubule estimated by Ferro et al. Therefore the following statement by the authors regarding the registration is not proven experimentally: "The binding site and register of the MAP7 MTBD helix on the MTs also matches that obtained from 3D reconstruction of MAP7 MTBD bound to Taxol stabilized-MTs18, demonstrating that MAP7 interacts with MTs in the same way independently of how the MTs are stabilized (Supplementary Fig. 3d)".

As it is, the cryo-EM data presented in this paper does not add information to the cryo-EM microtubule-MAP7-MTBD complex structure recently reported by Ferro et al. (Science 2022, ref 18). The resolution of the MTBD in the Ferro et al. reconstruction with resolved side chains also appears better than in this work.

Response 23): The reviewer is presumably referring to the reconstruction depicted in Fig. 1 and Fig. S3B of Ferro et al that was determined using the FL-MAP7, which has a higher MT binding affinity than MAP7 MTBD (K_D of 111 ± 12 nM for FL-MAP7 vs K_D of $0.94 \mu\text{M} \pm 0.73 \mu\text{M}$ for MAP7 MTBD). This higher affinity very likely contributes to greater occupancy of MAP7 on the MTs and ultimately for the quality of the MAP7 density presented in that study, albeit with the structure providing no information about how regions outside the MTBD enhance MT binding. Consistent with this interpretation, the density that Ferro et al present for a MAP7 MTBD construct (residues 60-170) in a co-complex with tau (Supplementary Fig. 6) is at lower apparent resolution both than their FL-MAP7 reconstruction and our MTBD reconstruction.

However, we accept that the previous presentation of our MAP7-MTBD-MT reconstruction was not as clear as it should have been. We have now made a number of additions and adjustments to improve this, including collecting more data for the MAP7-MTBD-MT reconstruction (39266 compared to 15857 “particles”, Supplementary Fig. 3a). Further, we have also carried out symmetry expansion, particle subtraction and refined a single protofilament as was done in Ferro et al and previously described in Debs et al (PMID: 32636254) (Supplementary Fig. 3a). This has improved the quality of the density attributable to MAP7 and its helical nature is more clearly visible (Fig. 2b). However, as noted above and in the text, the ability to visualize specific side chains in our MAP7-MTBD density is confounded by the averaging that occurs across tubulin dimers, which also seems to be a feature of the analysis conducted by Ferro et al (Supplementary Fig. 4c). The model fitting and analysis in the Ferro et al paper is carried out assuming the fragment of 87-134 residues of MAP7 MTBD as being the conserved MT binding fragment. Rendering of their deposited density (EMDB-25120 and 25119) for MAP7-FL and MAP7 MTBD+tau bound MTs in an equivalent way to our newly improved EM density reveals approximately similar density quality (Supplementary Fig. 6). As noted in Response 16, binding of the 110-residue long MAP7 MTBD across more than one dimer using cryo-EM data can only be inferred indirectly from modelling experiments, using Rosetta in the case of Ferro et al and AlphaFold2 in our analysis. As noted elsewhere, one of the novelties of our study is that our NMR experiments provide direct experimental data for MT binding by the extended MAP7 MTBD, thereby enhancing the modelling information.

We also agree that, as is commonly observed in MT-binding partner reconstructions, MAP7 is visualized at lower overall resolution than the MT itself. This is now presented more explicitly in a local resolution depiction of our updated MAP7-MTBD reconstruction (Supplementary Fig. 4b), and we have adjusted the text and data presentation accordingly.

[9] 4) When the authors refer to the symmetrized reconstruction it is not fully clear what it is the extent of the symmetrization. Are all tubulin dimers and bound MTBD averaged or all tubulin monomers (alpha and beta) with bound MTBD? The second case would occur if the presence of seams in the microtubule images were disregarded or if their location in all

microtubule images could not be accurately determined. If the claim is the former as suggested by the EM methods paragraph, then some evidence should be presented showing that indeed the alpha and beta tubulin structures are separated and that they show distinct structural features corresponding to their slightly different amino acid sequences. Note that averaging all monomers either by design or by accident may partially account for the poor resolution of the MTBD in the reconstruction. In any case the averaging procedure used was not adequate given that the proposed MTBD model (Fig. 7) extends almost completely over two tubulin dimers.

Response 24): We apologize about the lack of clarity in our processing pipeline and have now included more information in the methods text and figures about this (Methods section, lines 541- 557 and Supplementary Fig. 3). However, in summary, our ability to differentiate structural features of alpha- and beta- tubulin (Supplementary Fig. 3) provides good evidence that our reconstruction arises from averaging of tubulin dimers and not tubulin monomers, and are able to take into account the presence of the seam during symmetrization. However, as noted in Response 16, and reviewed by Nogales and Kellogg (PMID: 37702417), an important limitation of all cryo-EM studies of MAP7 to date is that reconstructions procedures involve averaging across 80 Å tubulin dimers, which results in blurring of information about binding of the 130 Å MTBD. Thus, the NMR data in our manuscript provides crucial additional information.

[11] 5) The fitting of the Alphafold model to the cryo-EM structure is poor. In Fig 2b the upper part of the MTBD model is completely out of the cryo-EM density. This is under-emphasized, and the authors do not show a zoom-in view of their Alphafold MAP7 model in their cryo-EM density (the MAP7 model is not shown in Fig. 2c for instance). A closer view of the alpha fold model fitted in Fig. 2b is provided in Supplementary Fig. 3.d but without a cryo-EM density overlay. Additionally, the alpha fold model also deviates substantially from the experimental model from Ferro et al in the part of MAP7 nearer tubulin alpha-3 (Supplementary Fig 3d). The AphaFold model could have been useful to support the fact that the new interaction the authors propose would be possible structurally, however it appear it is not the case from the above considerations.

Response 25): As noted in Response 7, the previous presentation of our model compared to our cryo-EM density was not clear. We apologize for this lack of clarity. We have redone this part of our analysis entirely, have added more data and rewritten much of the relevant text (Methods, Results and Discussion) supported by new Figures (Supplementary Fig. 5).

[12] 6) The authors indicate on page 15: "The register is supported by the observation of more flexible residues in the clefts between the tubulin subunits" However this is only partially true. For example, according to the Ferro et al. model MTBD residue R106 is at the middle of the gap between tubulin subunits yet this residue and the adjacent ones are shown to be 'rigid'. Overall, the otherwise interesting differences of dynamics obtained from the NMR experiments are not sufficiently clear to assign a precise registration of MAP7 and unfortunately this information cannot be derived from the cryo-EM data presented in this paper either.

Response 26): We agree with the reviewer that further studies would be needed to exactly define the register. For this reason we have modified the statements on lines 414-423.

[13] 7) In the summary structural model shown in bottom left panel of Figure 7 where two tubulin dimers are shown, some residues of the MTBD should be indicated to better relate it to areas mentioned in the text and other figures and to compare with other models. For example, does the segment between MTBD residues 86-139 is placed on the microtubule as in the Ferro et al. model? If yes, this should be acknowledged. The Figure title says 'based on the cryo-EM' but does not specify which cryo-EM data.

Response 27): The segment between MAP7 MTBD residues 63 to 151 in Figure 7 are based on the deposited model (PDB ID 8P6R) derived from cryo-EM experiments done in the course of this study. This has now been indicated in the figure legend.

[14] 8) Regarding the solution-state NMR experiment with the tubulin tails mimics, the authors mention "For the experiments the peptides were solubilized in NMR buffer and their pH adapted." The last statement about the pH is somewhat ambiguous: was the final buffer and pH of the 2 solutions used in the titration the same?

Response 28): Yes

[15] 9) The authors indicate on Page 5: "Further, extension of density for both α - and β -tubulin's H12 helix could be observed in the reconstruction and additional C-terminal residues Val437-Ser439 for α -tubulin and residues Ala428-Thr429 for β -tubulin were modelled into the density (Table 2). The ordering of these usually disordered residues could be explained by an interaction between the tubulin CTTs and MAP7; however, the direct interaction is not visible in our reconstruction and is presumably less ordered (Fig. 2c)."

This statement on the stabilization of H12 and C-terminal residues is not supported by the data provided in the paper. One would need to compare the densities of the residues mentioned here with the ones present in a naked microtubule obtained at a similar resolution.

Response 29): We did not present the data in support of these statements sufficiently clearly in the previous version of the manuscript but have now updated Fig. 2c and associated text to more clearly show how short extensions of the C-termini of tubulin are visualized in the MAP7-MTBD reconstructions. From this, we infer that MAP7 might interact with them albeit flexibly.

[16] 10) "the corresponding atomic model for the tubulin dimer is available in the PDB with accession number 8P6R." On a tubulin dimer only? The full model of the MTBD bound to two tubulin dimers (i.e., the model shown in Fig. 7) is what it needs to be deposited.

Response 30): We apologize for this lack of clarity. Although our data provide robust evidence about the MAP7 residues involved in binding across four tubulin monomers, as discussed above, the register of the helix cannot be unambiguously assigned from our data and hence we did not deposit a complete model of two tubulin dimers + MAP7-MTBD. Following improvements in our reconstruction and model building, our modified deposited model (PDB ID 8P6R) now consists of four tubulin monomers, including the central tubulin heterodimer

and the helical backbone (poly-Ala) of MAP7 MTBD, and we have now explicitly stated that this is what our PDB deposition corresponds to (See Methods, lines 565-583 and Data availability, lines 677-683).

REVIEWERS' COMMENTS

Reviewer #2 (Remarks to the Author):

The manuscript by Alder et al. contains high-quality experimental data but some of them were not presented clearly in the originally submitted version and in some cases novelty of the results was not obvious or explained sufficiently.

In the revised version, the authors carefully addressed all reviewers' questions and modified the manuscript substantially. This applies especially to the cryo-EM data that is now presented in more detail and discussed more clearly. Considering this improvement, I recommend the article for publication.

Reviewer #3 (Remarks to the Author):

In this revised version of the paper, the authors have carefully and satisfactorily addressed all my previous comments. I have no further issues to be addressed, and therefore, I recommend its publication in Nature Communications.